# Does Graph Prompt Work? A Data Operation Perspective with Theoretical Analysis

## Abstract

In recent years, graph prompting has emerged as a promising research direction, enabling the learning of additional tokens or subgraphs appended to the original graphs without requiring retraining of pre-trained graph models across various applications. This novel paradigm, shifting from the traditional "pre-training and fine-tuning" to "pre-training and prompting" has shown significant empirical success in simulating graph data operations, with applications ranging from recommendation systems to biological networks and graph transferring. However, despite its potential, the theoretical underpinnings of graph prompting remain underexplored, raising critical questions about its fundamental effectiveness. The lack of rigorous theoretical proof of why and how much it works is more like a "dark cloud" over the graph prompt area to go further. To fill this gap, this paper introduces a theoretical framework that rigorously analyzes graph prompting from a data operation perspective. Our contributions are threefold: **First**, we provide a formal guarantee theorem, demonstrating graph prompts' capacity to approximate graph transformation operators, effectively linking upstream and downstream tasks. **Second**, we derive upper bounds on the error of these data operations by graph prompts for a single graph and extend this discussion to batches of graphs, which are common in graph model training. **Third**, we analyze the distribution of data operation errors, extending our theoretical findings from linear graph aggregations (e.g., GCN) to non-linear graph aggregations (e.g., GAT). Extensive experiments support our theoretical results and confirm the practical implications of these guarantees.

## 1 Introduction

Graph Neural Networks (GNNs) have been widely used in analyzing various graph-structured data. A standard workflow using GNNs is the "pre-training and fine-tuning" paradigm, where a model is first trained on a large-scale, general-purpose dataset and then fine-tuned on a specific downstream task. While this method has been effective in transferring learned representations, it often needs many advanced tricks to retrain the model parameters for each new task, which can be computationally intensive and may not fully capture the unique characteristics of the downstream tasks, potentially limiting the model's generalization.

Inspired by the success of prompting techniques in natural language processing (NLP), there has been a growing interest in adapting similar ideas to graph data through "pre-training and prompting". Graph prompts (Sun et al., 2023b) modify the input graphs by adding learnable tokens or subgraphs, enabling the pre-trained GNN to better align with the requirements of downstream tasks without tuning the model parameters. Many empirical works (Sun et al., 2022; Liu et al., 2023; Tan et al., 2023; Huang et al., 2023; Ma et al., 2023) have found that graph prompting can reduce computational overhead, preserve the generality of the pre-trained model, and allow for seamless application across multiple tasks to achieve better expressive capability than the traditional paradigm.

Recently, some studies (Fang et al., 2024; Sun et al., 2023a) have realized that the reason why graph prompts work may relate to their capability in simulating various data operations like deleting/adding nodes/edges, changing node features, and even removing subgraphs. This makes graph prompts stand out from their counterpart in the NLP area and inspires many empirical applications like recommendation systems (Yang et al., 2023; 2024), biological networks (Diao et al., 2022),

transferring knowledge across different graph domains (Guo et al., 2023; Zhu et al., 2024b), and more. Unfortunately, despite these promising results, the theoretical basis of graph prompting remains underexplored. Existing works primarily rely on empirical validation and lack rigorous theoretical analysis to explain why graph prompts are effective and how they can be systematically designed. This gap is just like a "dark could" over the graph prompt area, raising critical questions about their broader applications and the development of more advanced methods that could leverage their full potential.

In light of these limitations, this paper provides a comprehensive theoretical framework for graph prompting from a data operation perspective. **First**, we establish rigorous guarantee theorems that demonstrate the underlying reason why graph prompts work is their capacity to simulate various graph data operations, and our theorems further answer why such capacity can make the pre-trained model meet new task requirements without retraining. **Second**, we derive upper bounds on the error introduced by graph prompts when simulating these data operations. We analyze this error for individual graphs and extend our discussion to batches of graphs, which is crucial for understanding the scalability and generalization of graph prompts in practical scenarios where models are usually trained on multiple graphs. **Third**, we explore the distribution of the data operation error and extend our theoretical findings from linear graph aggregations, such as Graph Convolutional Networks (GCNs), to non-linear aggregations like Graph Attention Networks (GATs). This extension demonstrates the robustness of our theoretical framework across different GNN architectures and provides insights into how non-linearity affects the effectiveness of graph prompts. We conduct extensive experiments to confirm our theoretical findings. By offering such a solid theoretical foundation for graph prompting, our work advances the understanding of how and why graph prompts work, guides for designing more effective prompting techniques, and empowers researchers and practitioners to leverage them with greater confidence in various applications.

## 2 BACKGROUND

**Graph Prompt.** Compared with "pre-training and fine-tuning", which first trains a graph model via some easily accessible task on the graph dataset and then tries to adapt the model to a new task (or even a new graph dataset), "pre-training and prompting" aims to keep the pre-trained model unchanged but adjust the input data to make the downstream task compatible with the pre-training task. Mathematically, let $F_{\theta^*}$ be a graph model where its parameters ($\theta^*$) have been pre-trained and frozen; $T_{dow}$ be the downstream task, in which the task objective is measured by a loss function $\mathcal{L}_{T_{dow}}$. Let $\mathcal{G}$ be a graph dataset and each graph instance $G \in \mathcal{G}$ can be denoted as $G = (\mathcal{V}, \mathcal{E}, \mathbf{X}, \mathbf{A})$ where $\mathcal{V}$ denotes the node set with a node feature matrix $\mathbf{X} \in \mathbb{R}^{|\mathcal{V}| \times F}$; $\mathcal{E}$ denotes the edge set and the connection of nodes can be further indicated by the adjacent matrix $\mathbf{A} \in \{0, 1\}^{|\mathcal{V}| \times |\mathcal{V}|}$. Let $P_\omega$ denote a parameterized graph prompt function with learnable $\omega$. In most cases, graph prompts consist of some token vectors or subgraphs which will be integrated into the original graph $G$. $P_\omega$ indicates how to define such graph prompts and how to combine them with the original graph to generate a new graph: $G_\omega = P_\omega(G)$. Graph prompt learning aims to optimize the following target:

$$\omega^* = \arg\min_\omega \sum_{G \in \mathcal{G}} \mathcal{L}_{T_{dow}}(F_{\theta^*}(P_\omega(G)) \tag{1}$$

Without loss of generality, we assume all these tasks are graph level (e.g., graph classification). That means $F_\theta(G)$ will output a graph-level embedding for the downstream task. For node-level and edge-level tasks, many studies (Sun et al., 2023a; Liu et al., 2023) have proved that we can always find solutions to translate these tasks to the graph-level task.

**GPF and All-in-One.** Current graph prompt designs, as described in the review by Sun et al. (2023b), can be primarily categorized into two types: prompt as token vectors added to node features, and prompt as additional graph inserted to the original graph. In the rest of this paper, we focus on two representative frameworks: GPF (Fang et al., 2022) as an example of adding extra prompt vectors, and All-in-One (Sun et al., 2023a) as an example of adding prompt subgraphs. Our choice is motivated by our interest in simulating graph operations at a theoretical level. The GPF and All-in-One frameworks provide the most fundamental approaches among these methods. The rest graph prompt designs can usually be treated as their special cases or natural extensions (Sun et al., 2023b).

Specifically, GPF aims to learn a token vector $\omega \in \mathbb{R}^{1 \times F}$ where $F$ is the same dimension of the node features in the original graph, then the prompt token is directly added to each node's feature vector, making the original feature matrix $\mathbf{X} = \{x_1, \cdots, x_N\}$ changed to $\mathbf{X}_\omega = \{x_1 + \omega, \cdots, x_N + \omega\}$. In this way, the original graph $G = (\mathcal{V}, \mathcal{E}, \mathbf{X}, \mathbf{A})$ is changed to $P_\omega(G) = G_\omega = (\mathcal{V}, \mathcal{E}, \mathbf{X}_\omega, \mathbf{A})$.

All-in-One offers the prompt as a graph format by defining prompt tokens, token structures and inserting patterns. Let $\Omega \in \mathbb{R}^{k \times F}$ be the learnable matrix corresponding to $K$ prompt tokens. Let $\mathbf{A}_{in} \in \{0, 1\}^{k \times k}$ indicate token structures where $A_{ij} = 1$ means there is an inner link connecting the $i$-th and the $j$-th tokens and vice versa. $A_{in}$ can be calculated by the inner product of these tokens. Similarly, the inserting pattern tells us how to connect each token to the original graph nodes, which is denoted by a cross matrix $\mathbf{A}_{cro} \in \{0, 1\}^{k \times N}$. Then the graph prompt changes the original graph $G$ to $G_\omega = (\mathcal{V}_\omega, \mathcal{E}_\omega, \mathbf{X}, \mathbf{\Omega}, \mathbf{A}_\omega)$ where $\mathcal{V}_\omega$ is the collection of the original nodes and the token nodes; $\mathcal{E}_\omega$ includes the original edges, inner links among tokens and the cross links between tokens and the original nodes; $\mathbf{A}_\omega$ is the collection of $\mathbf{A}, \mathbf{A}_{in}$, and $\mathbf{A}_{cro}$.

**Motivations.** Initially, Fang et al. (2022) have proved that graph prompt can simulate any graph data operation (e.g., deleting/adding nodes/edges, changing node features, or removing subgraphs, etc). That means for any graph data operation $t(\cdot)$, we can always learn a graph prompt reaching $F_{\theta^*}(G_\omega) = F_{\theta^*}(t(G))$. However, this equivalence needs a very strong precondition: the graph model $F$ should not contain any non-linear layer, which is apparently very hard to meet in the practical solutions. Later, Sun et al. (2023a) extended this finding with more advanced prompts and empirically observed that the error to such approximation $F_{\theta^*}(G_\omega) \rightarrow F_{\theta^*}(t(G))$ may relate to the non-linear layers of the graph model and the prompt design. However, these observations are not followed by a critical theory proof. Recently, there has been an increasing number of empirical works on graph prompts achieving success in various applications. Unfortunately, the theoretical basis of graph prompt is still very tumbledown and we still have not figured out why graph prompts work in theory, especially for questions like *how powerful the graph prompt is in manipulating graph data?* and *why such capability works for downstream tasks?*

In this paper, we go deeper in theory for the graph prompt capability of manipulating data. We conduct a comprehensive effectiveness analysis of general graph prompt learning through the concepts of "***bridge sets***" and "***$\epsilon$-extended bridge sets***" (see in section 3.3). Based on extensive theoretical derivations and substantial experimental evidence, we establish theorems related to the error bound of graph prompts in simulating graph data operations. Our goal is to figure out in theory how this error changes from a single graph to a batch of graphs, from a linear model to a non-linear model, and which factors relate to this error. Through this work, we wish to push forward the graph prompt area with a more solid theory basis, help researchers to design more scientific graph prompt techniques, and offer them theory confidence for their further usage.

# 3 WHY GRAPH PROMPT WORKS? A DATA OPERATION PERSPECTIVE

Let $F_{\theta^*}$ be any given GNN model that has been pre-trained on a given task $T_{pre}$. Here $\theta^*$ means the parameters have been determined and frozen. For a given graph instance $G_{ori}$, we can expect the model to output appropriate graph-level embedding on $T_{pre}$ because this model has already been trained on this task. However, when we try to use this model on a new task $T_{dow}$, the output embedding, $F_{\theta^*}(G_{ori})$, can not guarantee acceptable performance because the pre-training task $T_{pre}$ may be incompatible with downstream tasks $T_{dow}$.

## 3.1 PERSPECTIVE FROM MODEL TUNING

To fill this gap, "pre-training and fine-tuning" aims to adapt the pre-trained model to a new version and wish it could perform better. Assume there exists an optimal function, say $C$, which can map $G_{ori}$ to the embedding $C(G_{ori})$ to achieve good performance on $T_{dow}$. The nature of "pre-training and fine-tuning" is to hope the fine-tuned graph model could approximate to $C(G_{ori})$:

$$F_{\theta^* \rightarrow \theta^\#}(G_{ori}) \rightarrow C(G_{ori}) \tag{2}$$

However, achieving this goal usually requires fine-tuning the graph model, which is not always efficient and needs many empirical tuning tricks. The tuning course may be even harder if the pre-trained model is ill-designed for the downstream task. In addition, we can not guarantee that

fine-tuning the pre-trained model (i.e. $\theta^* \to \theta^\#$) can always surpass training the model from scratch because the preserved knowledge may contribute negatively to the downstream task.

## 3.2 PERSPECTIVE FROM DATA OPERATION

Instead of the above model-level tricks, graph prompts provide a data-level alternative. Some prior works (Fang et al., 2024; Sun et al., 2023a) have initially proved that graph prompts can simulate any graph operations (e.g., deleting/adding nodes/edges/subgraphs, changing node features, etc). However, how effective of graph prompt is and why this works for the new task are still not yet answered. To answer these questions, we first explain our data operation perspective by a theorem as follows:

**Theorem 1.** *Let $F_{\theta^*}$ be a GNN model pre-trained on task $T_{pre}$ with frozen parameters ($\theta^*$); let $T_{dow}$ be the downstream task and $C$ is an optimal function to $T_{dow}$. Given any graph $G_{ori}$, $C(G_{ori})$ denotes the optimal embedding vector to the downstream task (i.e. can be parsed to yield correct results for $G_{ori}$ in the downstream task), then there always exists a bridge graph $G_{bri}$ such that $F_{\theta^*}(G_{bri}) = C(G_{ori})$.*

A detailed proof of Theorem 1 can be seen in Appendix A.3.1, from which we can find that for any given graph $G_{ori}$, there always exists a bridge graph, say $G_{bri}$, making the following equation hold:

$$F_{\theta^*}(G_{bri}) = C(G_{ori}) \tag{3}$$

That means, without needing to tune the model, we can try to find a data operation method that translates $G_{ori}$ to $G_{bri}$ with the pre-trained model unchanged. Graph prompts can be treated as a learnable data operation framework to help us manipulate these graph data. In this way, we can significantly reduce the difficulty of traditional fine-tuning work, improve the performance on a new task (or even a new dataset), and further enhance the generalization of graph neural networks. With this perspective, our next question is: *How difficult to find such a bridge graph using graph prompts?*

## 3.3 MEASURING THE DIFFICULTY OF FINDING BRIDGE GRAPHS VIA GRAPH PROMPTS

Graph prompts can be viewed as a type of graph transformation operator. For example, the simplest graph prompt is just adding a specific prompt token vector $p_\omega$ to each node feature of the graph and then we can transform this graph $G$ into a family of graphs $\{P_\omega(G)|\omega \in \mathbb{R}^F\}$, where $P_\omega(G)$ represents the output graph obtained by graph prompt on $G$. Once $\omega$ is determined, the corresponding data transformation rule and unique output graph data are also defined. This family can be understood as the "***transformation space***" of graph $G$ under prompt $P$, denoted as $D_P(G)$. If the prompt operator maps the original graph $G_{ori}$ to a bridge graph $G_{bri}$ (i.e., $P(G_{ori}) = G_{bri}$), then applying the pre-trained model $F_{\theta^*}$ yields $F_{\theta^*}(P(G_{ori})) = F_{\theta^*}(G_{bri})$, which conforms to the downstream task. In this process, we achieve seamless alignment of upstream and downstream tasks solely through data transformation operators, without relying on tuning the model's parameters.

**Definition 1** (**Bridge Set and $\epsilon$-extended Bridge Set**). *The bridge set of a graph $G$ is defined as:*

$$B_G = \{G_p \mid F_{\theta^*}(G_p) = C(G)\}$$

*where $F_{\theta^*}$ is the frozen graph model from the pre-training task, and $C$ is the optimal function for the downstream task. The $\epsilon$-extended bridge set of $G$ is a relaxed version of the bridge set, which is defined as:*

$$\epsilon\text{-}B_G = \{G_p \mid \epsilon = \|F_{\theta^*}(G_p) - C(G)\| \le \epsilon^*\}.$$

Achieving a transformation exactly equivalent to $C(G)$ is highly non-parametric and nonlinear. Finding the corresponding $G_{bri}$ or even the bridge set for any $G_{ori}$ involves solving complex, high-order nonlinear equations. This task becomes virtually impossible manually, especially if the pre-training method integrates multiple tasks or intricate mechanisms. Fortunately, graph prompts $P_\omega$ can be viewed as parameterized fitting for these operators and they usually contain very lightweight parameters, which dramatically simplify the search space compared with manually designed strategies. The effectiveness of graph prompting methods hinges on their ability to approximate these operators closely—whether they can uniformly project $G$ in the dataset into the bridge set $B_G$, or at least map them into the extended bridge set with a small upper error bound $\epsilon^*$.

# 4 THE UPPER BOUND OF DATA OPERATION ERROR VIA GRAPH PROMPT

## 4.1 UPPER BOUND OF THE ERROR ON A SINGLE GRAPH

Here we aim to demonstrate that using graph prompts provided by frameworks such as GPF and All-in-One, denoted as parameterized operators $P_\omega$ with parameter $\omega$, can consistently project $G$ into an $\epsilon$-extended $B_G$, where $\epsilon$ has a uniform upper bound. This would initially validate the effectiveness of graph prompting methods in leveraging the potential of pre-trained models without compromising their expressive power. If $G$ can be seamlessly projected into $B_G$ or an $\epsilon$-extended $B_G$ (for small $\epsilon$), it would indicate excellent performance and full utilization of the model's capabilities.

To this end, we first conduct a quantitative analysis of $P_\omega$'s graph transformation approximation ability on a single graph. With our proposed data operation perspective, we can reformulate the findings in Fang et al. (2022) as follows:

**Theorem 2.** *Given a GPF-like prompt vector $p_\omega$, if a GCN model $F_\theta$ does not have any non-linear transformations, then there exists an optimal $\omega$ for any input graph $G$ such that $P_\omega(G) \in B_G$.*

This theorem is proved by Fang et al. (2022) but it's important to note that all GNN models employ non-linear transformations. According to the function approximation theorem for neural networks (Hornik et al., 1989), the core of improving a model's approximation and simulation ability lies in its non-linear components. Removing these non-linear parts would limit the model to approximating only linear transformations and functions. To demonstrate the effectiveness of graph prompt learning in real downstream tasks, we offer the following theorems further:

**Theorem 3.** *Given a GPF-like prompt vector $p_\omega$, if a GCN model $F_\theta$ has non-linear function layers but the model's weight matrix is row full-rank, then there exists an optimal $\omega$ for any input graph $G$ such that $P_\omega(G) \in B_G$.*

**Theorem 4.** *Given the All-in-One-like prompt graph $SG_\omega$ (a subgraph containing prompt tokens and token structures), if a GCN model $F_\theta$ does not have any non-linear transformations, or has non-linear layers but the model's weight matrix is row full-rank, then there exists an optimal $\omega$ for any input graph $G$ such that $P_\omega(G) \in B_G$.*

Theorems 3 and 4 are proved in Appendix A.3.2. Although we've only added the row full-rank condition, these two theorems significantly expand the applicability of Theorem 2. According to Pennington & Worah (2017), well-trained models mostly contain full-rank matrices, which can be easily guaranteed by some tricks like orthogonal initialization, He initialization, etc. Raghu et al. (2017) also find that a full-rank parameter matrix in the model usually indicates stronger expressiveness. Intuitively, a weight matrix in the graph model usually indicates how to project the input graph into some latent embedding for the downstream task. According to the basic knowledge of linear algebra, when the weight matrix is row full-rank, we can always restore the input from the output. That means we can always find an appropriate input format to meet various downstream requirements. Therefore, from a practical empirical perspective, we can assume that in most cases, both GPF-like and All-in-One-like frameworks can achieve seamless projection of $G$ into $B_G$, demonstrating the effectiveness and rationality of graph prompting.

For the cases where weight matrices are not full-rank, we have found that the error value ($\epsilon$) of the extended bridge set ($\epsilon$-$B_G$), into which the prompting framework can map $G$, is positively correlated with the distance of the graph model parameter matrix from being full-rank. However, a consistent upper bound does exist for the error $\epsilon$ of the extended bridge set when the matrix rank is determined. This means that even in some extreme cases, graph prompt learning can still guarantee a certain level of performance without experiencing extremely unexpectedly poor results:

**Theorem 5.** *For a GCN model $F_\theta$, assume at least one layer's parameter matrix is not full rank, for GPF or All-in-One prompt, there exists an upper bound of $\epsilon$ such that for any input graph $G$, there exists an optimal $\omega$ where $P_\omega(G) \in \epsilon$-$B_G$, with $\epsilon \leq \zeta(\theta^*) \cdot \kappa(G)$, where $\zeta(\theta^*)$ is an implict function correspond to the model and $\kappa(G)$ is an implic function corresponding graph G, denoting the two parts of the error boundaries.*

Theorem 5 is proved in Appendix A.3.3, where the upper bound of the error $\epsilon$ can be further expressed as follows:

$$\zeta(\theta^*)\kappa(G) = \sin(\Phi/2)\|C(G)\| \tag{4}$$

The upper bound has two terms, one part is related to the model because $\sin(\Phi/2)$ can be treated as a measurement of the model's expressiveness, the details of which can be seen in Appendix A.3.3. The other part is related to $\|C(G)\|$. Since the graph prompt aims to approximate $G$ with $P_\omega(G_{ori})$, that means the error is also related to the prompt design. In this way, we in theory confirm the intuitional findings by Sun et al. (2023a) as mentioned in our motivation section.

The upper bound of $\epsilon$ given in Theorem 5 reveals the potential distortion of $B_G$'s shape when the matrix is not full-rank and the model's expressive power is insufficient. This can lead to an increased distance between $B_G$ and the transformation domain $D_P(G)$ of GPF or All-in-One prompts. To confirm this judgment, we conducted a quantitative analysis using numerical methods for the case of non-full rank matrices. We make the graph model weight matrix be not full-rank (with rank $n-r$ where $n$ is the full rank number and $r = 0, \cdots n$) and use the distance between the embedding vectors of $P_\omega(G)$ and $C(G)$ as the loss function. We applied stochastic gradient descent with a learning rate of $0.0001$ to optimize $\omega$ until convergence. To avoid local optima, we repeated the process multiple times

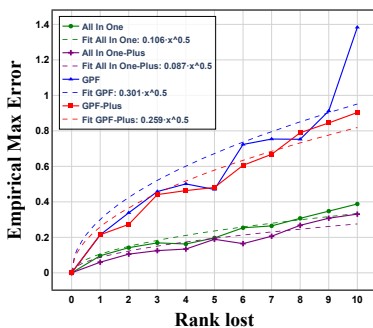

Figure 1: Error w.r.t matrices rank.

with different initializations for each graph. The results are shown in Figure 1, where the vertical axis is the empirical max $\epsilon$ of the extended $B_G$ and the horizontal axis is $r$ where $r = 0, 1, 2, \cdots, 9, 10$, making the matrix rank become $n - r$. Here GPF-Plus contains multiple tokens, and All-in-One-Plus treats the inserting pattern as learnable weights. We can find that with the increase of $r$, the rank of the model matrix becomes lower, making the model expressiveness worse and then leading to a larger error bound. Besides, more advanced graph prompts (e.g., GPF-Plus, All-in-One, and All-in-One-Plus) generally have a lower bound than the naive one (e.g., GPF).

## 4.2 EXTEND THE ERROR BOUND DISCUSSION TO A BATCH OF GRAPHS

In Sections 3 and 4.1, we have proved that graph prompting frameworks can indeed fit graph transformation operators given a single graph, thereby exploiting model capabilities. However, in other cases, we often train the model via a batch of graphs and seek to find better performance over the whole graph dataset. Correspondingly, we should aspire to transform each graph $G$ in the downstream dataset into its corresponding $B_G$ or $\epsilon$-$B_G$ (for small $\epsilon$). If such a uniform upper bound $\epsilon^*$ exists, it would theoretically validate the excellent performance of graph prompting in general downstream tasks, confirming the rational utilization of powerful upstream models.

For a batch of graphs, the complexity and information contained in the graph prompt become particularly important. For instance, the increased number of prompt vectors in GPF (a.k.a GPF-Plus) and the selection of a larger size of the prompt graph in All-in-One greatly expand the transformation space of graph $G$ under prompt $P$ (see $D_P(\cdot)$ in section 3.3). A larger transformation space corresponds to a smaller $\epsilon$ upper bound. In our theoretical analysis, we found that when the prompt takes an overly simple form, the capability of prompt learning is limited. This manifests as a theoretical lower bound of the bridge set extension as suggested in Theorem 6:

**Theorem 6.** *For a GCN model $F_\theta$, for GPF with a single prompt vector or All-in-One with a single-token graph prompt, given a batch of graphs $\mathcal{G} = \{G_1, \cdots, G_i, \cdots, G_n\}$, the root mean squared error (RMSE) over $\{\epsilon_1, \cdots, \epsilon_n\}$ has a lower bound $\epsilon^o$ such that $RMSE(\epsilon_1, \cdots, \epsilon_n) \geq \epsilon^o$.*

Theorem 6 is proved in Appendix A.3.4 and we also give the detailed formulation of $\epsilon^o$ in the proof, from which we can find that $\epsilon^o$ is related to graph data and the prompt token. This indicates that when the downstream task dataset is relatively large, we must correspondingly increase the transformation space of the prompt to better utilize the model's capabilities, which also aligns with existing empirical observations (Liu et al., 2021). Then our next question is: *With the increase of graphs, how does graph prompt complexity increase with their error bound increased?*

Intuitively, a good graph prompt should not increase its complexity faster than the growth of the dataset because in that case the effectiveness of prompt learning in practical applications would be significantly compromised. Fortunately, we found that for relatively large datasets, the scale of prompts required for prompt learning is highly controllable with the number of needed tokens for the prompt almost constant, far from the increase of graphs. This explains why even with relatively

large downstream datasets, as reflected in Sun et al. (2023a), empirical results using medium-scale prompts can still achieve excellent outcomes. Compared to fine-tuning the entire model parameters, Theorem 7 (see the proof in Appendix A.3.4) indicates why graph prompting can achieve comparable or even better results with less parameter adjustment scale:

**Theorem 7.** *Given a GCN model $F_\theta$, an All-in-One-like graph prompt with multiple prompt tokens, and a dataset $\mathcal{G}$ with $M$ graphs, there exists an upper bound denoted by $\epsilon^*$, making an optimal $P_\omega$ such that $\forall G_i \in \mathcal{G}, P_\omega(G_i) \in \epsilon_i\text{-}B_{G_i}$, and $\sqrt{\sum_{i=1}^{M} \epsilon_i^2/M} \leq \epsilon^*$. $\epsilon^*$ can be further calculated as follows:*

$$\epsilon^* = \sqrt{\sum_{i=k+1}^{M} \lambda_i/M} \tag{5}$$

Here for the $M$ graphs in $\mathcal{G}$, we first construct an optimal solution matrix according to function $C$, thus have: $S = [C(G_1), \ldots, C(G_M)]$. Then $V = S^\top S \in \mathbb{R}^{M \times M}$ denotes the correlation matrix of downstream solutions upon such graph dataset. The eigenvalues of $V$ sorted by the descending order can be denoted as $\{\lambda_1, \cdots, \lambda_M\}$. Then the upper bound $\epsilon^*$ can be treated as the mean square over the smallest $M - k$ eigenvalues. In practice, the eigenvalues of $V$ in datasets often exhibit an exponential decay (Johnstone, 2001). This explains the rapid decrease in error rate as $k$ increases, proving that prompts are not only effective but also efficient. With the increasing number of graphs $M$ in the dataset, the largest $k$ ($k \ll M$) eigenvalues can almost explain most of the matrix, which means using small-scaled prompt tokens can achieve reasonably accurate results. This finding is also consistent with many existing empirical researches (Liu et al., 2023; Sun et al., 2023a; Wang et al., 2024; Zhu et al., 2024a).

### 4.3 VALUE DISTRIBUTION OF THE DATA OPERATION ERROR WITH GRAPH PROMPT

In the previous sections, we established a theoretical upper bound of $\epsilon \leq \epsilon^*$, allowing the graph prompt $P_\omega$ to map a given graph $G$ into the $\epsilon$-extended $B_G$ range. However, the conditions to reach this upper bound are often difficult to meet, making the theoretical upper bound usually correspond to some corner cases which may be not that practical in processing empirical experimental analysis. To offer stronger practical guidance for researchers' general experimental purposes, the value distribution of $\epsilon$ (a.k.a "error range"), comes to our next point of interest. The error range analysis indicates a quantitative degree of the bridge set extension, which includes estimating the mean, and variance, and finding the approximate distribution pattern.

**Definition 2** (**Graph Embedding Residual Vector**). *Consider a graph model with Leaky-Relu as their activation function, we denote the graph embedding residual vector as $\boldsymbol{\beta} \in \mathbb{R}^{1 \times F}$ where $F$ is the graph embedding dimensions. Each entry of $\boldsymbol{\beta}$ is related to the bridge graph embedding, graph prompt, and the original graph, which is mathematically defined as follows:*

$$[\boldsymbol{\beta}]_j = \sum_i -\alpha^k w_i v_{ij} \tag{6}$$

where $\alpha$ is the parameter of Leaky-Relu; $v_{ij}$ is the $j$-th element in the $i$-th node embedding of the bridge graph $G_{bri}$; $w_i$ is the weight of the $i$-th node for graph pooling (e.g., summation graph pooling means for any node $i$ in $G_{bri}$, we have $w_i = 1$); $k$ can be either 0 or $-1$ and depends on the original graph, graph prompt, and the bridge graph, the details of which can be seen in Appendix A.3.5. Here, we assume that $\boldsymbol{\beta}$ should follow an i.i.d. normal distribution with mean 0 and variance $c$ (we carefully

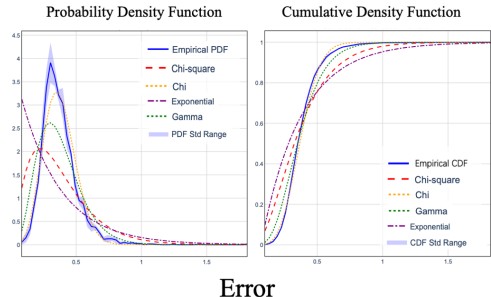

Figure 2: real $\epsilon$ distribution and fitted curves

discuss the rationality of this assumption in Appendix A.3.5): $\boldsymbol{\beta} \sim \mathcal{N}(0, cI_n)$ where $I_n$ is the $n \times n$ identity matrix and $c$ is a positive constant. Then we theoretically find that $\epsilon$ conforms to the Chi distribution ($\chi_k$):

**Theorem 8.** *Given a GCN model $F_\theta$ with the last layer parameter matrix having rank $F - r$ ($F$ is the graph embedding dimension, $r$ is the rank lost), an input graph $G$, for the optimal $\omega$, $P_\omega(G) \in \epsilon\text{-}B_G$. If the Graph Embedding Residuals follow the i.i.d. normal distribution, then $\epsilon$ follows a Chi distribution $\chi_r$ with $r$ free variables.*

We give the proof of Theorem 8 in Appendix A.3.5. In the practical settings, the distribution of graph residual terms may slightly diverge from i.i.d. normal distribution, making the real distribution of $\epsilon$ a little different from standard $\chi_k$. Besides, there is also a theoretical upper bound $\epsilon^*$, making the real distribution of $\epsilon$ more like a truncated $\chi_k$. The statistical measures of this distribution can be easily obtained as follows:

| Distribution | Notation | p-value |
|---|---|---|
| Chi | $\chi$ | 0.65 |
| Gamma | $\Gamma$ | 0.23 |
| Chi-squared | $\chi^2$ | 0.04 |
| Exponential | Exp | 0.01 |

Table 1: p-values w.r.t distributions

**Corollary 1** (Statistical Measures and Confidence Values of $\epsilon$). *The mean of $\epsilon$ is $c\sqrt{2}\frac{\Gamma((r+1)/2)}{\Gamma(r/2)}$, the variance is $c^2\left(r - 2\frac{[\Gamma((r+1)/2)]^2}{[\Gamma(r/2)]^2}\right)$, and confidence values can be obtained through $C_\chi^{r,p}$ using numerical methods or table lookup, where $c$ is the scaling factor compared to the standard distribution, and $r$ is the number of dimensions lost compared to a full-rank matrix.*

To further confirm our theoretical findings, we compare the real-world distribution of $\epsilon$ with 4 commonly used distribution patterns (Chi, Chi-square, Exponential, and Gamma). Figure 2 presents the fitting results and Table 1 shows the $p$-value significance, from which we can see that the Chi distribution provides the best approximation within a non-extreme range of $\epsilon$.

### 4.4 Extend the Discussion from Linear to Non-linear Aggregations

While our previous analysis focused on GCN or linear aggregation models that can be represented in the form of "diffusion matrices", many advanced models utilizing attention mechanisms exhibit distinctly different characteristics. Their aggregation methods involve the computation of attention matrices, which in turn depend on the node feature vectors of $G$. This can be considered as a non-linear model w.r.t $G$'s feature matrix. In our analysis, we use Graph Attention Networks (GAT) as an exemplar, as the attention mechanism in GAT is a common component in many non-linear models. Fortunately, the guarantees provided by our theorems do not differ significantly for these models. This indicates that even as models become more non-linear and complex, graph prompting can still effectively harness the powerful capabilities of pre-trained models:

**Theorem 9.** *Let $F_\theta$ be a GAT model. If any layer of the model has a full row rank parameter matrix, then for the All-in-One prompting framework, for any input graph $G$, there exists an optimal $\omega$ such that $P_\omega(G) \in B_G$. When the parameter matrix is not full rank, there is an upper bound $\mu(\theta) \cdot \lambda(G)$ making $P_\omega(G) \in \epsilon\text{-}B_G$, $\epsilon \leq \mu(\theta) \cdot \lambda(G)$.*

We give a detailed proof in Appendix A.3.6. The above theorem demonstrates the robustness of graph prompting methods across different types of GNN architectures, including those with non-linear attention mechanisms. The consistency of these results with our earlier findings for linear models suggests that the fundamental principles of graph prompting remain effective even as we move towards more complex and non-linear model architectures.

## 5 Experiments

### 5.1 Experimental Settings

**Data Preparation:** We first confirm our theoretical findings on synthetic datasets because these datasets offer controlled environments, allowing us to isolate specific variables and study their impacts. We generate these datasets by defining the dimension of graph feature vector ($F$), average of graph node numbers ($N_{avg}$), average of graph edge numbers ($E_{avg}$), and number of graphs in the dataset ($M$). These parameters characterize both individual graphs and the entire dataset, facilitating our study of the relationship between these features and $\epsilon$. We further conduct the experiments on the real-world dataset in Appendix B, from which we can find similar observations.

**Model Settings:** We utilize two GNN frameworks: GCN (representing linear models) and GAT (representing non-linear models). We limit our experiments to these two models as other models

follow similar patterns. Unless otherwise specified, we use a 3-layer GNN with Leaky-ReLU activation function and feature dimension $F = 25$. For full-rank matrix studies, we ensure each layer's matrix is full-rank (selected after pre-training). For non-full-rank matrix studies, we set the rank loss to 5 by default. The default ReadOut method is mean pooling.

**Training:** In the graph prompting training process, we perform gradient descent on the parameters $\omega$ of the graph prompt $P_\omega$ using the Adam optimizer. We use a learning rate of $1 \times 10^{-4}$ and weight decay of $5 \times 10^{-5}$. We implement an early stopping mechanism with a maximum of 2,000 epochs by default. Our loss function is defined as $\|F_{\theta^*}(G_p) - C(G)\|$ for single-graph tasks, and $\sqrt{\sum_{G \in \mathcal{G}} \|F_{\theta^*}(G_p) - C(G)\|^2 / M}$ for multi-graph tasks where $G_p$ is the combined graph with $G$ and graph prompt, and $C(G)$ means an optimal function to the downstream task, which is not accessible without a specific task. Since the ultimate purpose of graph prompting is to approximate graph operation, we here treat $C(\cdot)$ as various graph data permutations such as adding/deleting nodes, adding/deleting/changing edges, and transforming features of a given graph $G$. Then we wish to see how well the graph prompt reaches $C(G)$ by manipulating graph data with a graph prompt. For more detailed experimental settings, please check in the Appendix C. We open our testing code at https://anonymous.4open.science/r/dgpwadopwta/

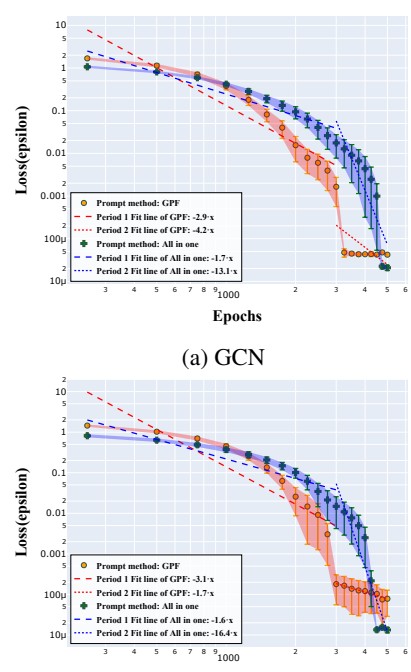

(a) GCN

(b) GAT

Figure 3: convergence rate analysis

## 5.2 ON MAPPING TO $B_G$ WITH SINGLE GRAPH

According to Theorems 3, 4, and 9, error-free projection can be achieved in full-rank situations. Here we investigate convergence properties with a maximum of 5,000 epochs. Figure 3 presents the results for GPF and All-in-One prompts with GCN and GAT, respectively. From the results we can find that for single-graph, full-rank matrix scenarios, both GPF and All-in-one approaches show loss converging to zero, which is consistent with our theoretical findings.

## 5.3 ON MAPPING TO $\epsilon$-$B_G$ WITH SINGLE GRAPH

Theorem 5 states that in non-full-rank situations, there exists an upper bound on the error. Here we extensively examine the relationship between various parameters and the error upper bounds in the context of non-full-rank matrices given a single graph. Since showing this upper bound in practice is usually intractable, we fix all other parameters and employ five pre-trained models. For each fixed pre-trained model, we conduct experiments and repeat each experiment 30 times. Then we take the maximum loss from these repetitions as the approximation to the upper bound.

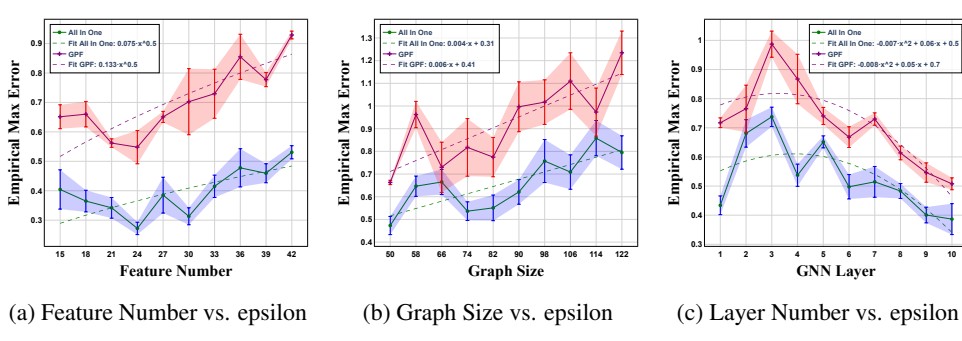

(a) Feature Number vs. epsilon    (b) Graph Size vs. epsilon    (c) Layer Number vs. epsilon

Figure 4: epsilon range analysis

Subsequently, we calculate the mean and standard deviation of these upper bounds to generate plots. The parameters of interest include rank loss (as shown in Figure 1 in section 4.1), node feature dimension (Figure 4a), graph size (Figure 4b), and model layer number (Figure 4c). Intuitively, with the increase of data complexity (e.g., larger features and graph size), the upper bound becomes larger in general. As the graph model becomes more complicated (e.g., layer number increase), the projected space becomes larger making the error bound intend to be smaller. When weight rank declines, the model's capacity intends to poor results, making the error bound increase. More advanced graph prompts (e.g., All-in-One) usually have a lower error bound than the naive one (e.g., GPF). These observations can be naturally inferred from our theoretical analysis in section 4.1.

## 5.4 ON MAPPING TO $\epsilon$-$B_G$ WITH MULTIPLE GRAPHS

Theorem 6 discusses a lower bound on the RMSE over the errors on multiple graphs with a single prompt token. In this section, we conducted experiments on the number of graphs in the dataset w.r.t the empirical minimum error. As shown in Figure 5a, the minimum error shows an upward trend and then tends to saturate, which is highly consistent with the findings in Theorem 6.

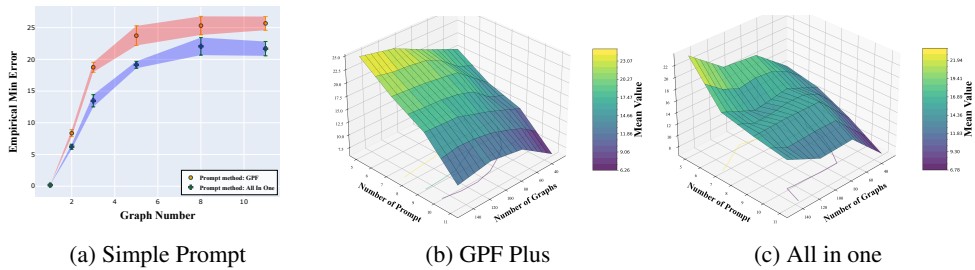

| (a) Simple Prompt | (b) GPF Plus | (c) All in one |

Figure 5: $\epsilon$ range based on multiple graphs analysis

Theorem 7 suggests that for the graph prompt with multiple tokens and multiple graphs, a small $k$ tokens is sufficient to achieve good performance. In particular, we wish to see how the error (loss) changes as $M$ (number of graphs) increases while $k$ remains fixed, and its counterpart case: how does the error change as $k$ increases while $M$ remains fixed? Here we explored the relationship between the number of prompt tokens, the number of graphs in the dataset, and the error. We present experimental results in Figure 5b and Figure 5c, which indicate two surfaces. From these figures we can find that both GPF and All-in-One show similar effects: when the number of prompt tokens exceeds 10, the error becomes relatively small. As the number of prompt tokens increases further, the loss does not significantly decrease. Similarly, when the number of graphs increases and the number of prompt tokens is large, the decrease in error is also not that obvious.

## 6 CONCLUSION

This paper addresses the theoretical gap in graph prompting by introducing a comprehensive framework from a data operation perspective. We introduced the concepts of "bridge sets" and "$\epsilon$-extended bridge sets" to formally demonstrate that graph prompts can approximate graph transformation operators, effectively bridging pre-trained models with downstream tasks without retraining. Our contributions are threefold: first, we established guarantee theorems confirming that graph prompts can simulate various graph data operations, explaining their effectiveness in aligning upstream and downstream tasks. Second, we derived upper bounds on the approximation errors introduced by graph prompts for both individual graphs and batches of graphs, highlighting how factors like model rank and prompt complexity influence these errors. Third, we analyzed the distribution of these errors and extended our theoretical findings from linear models like GCNs to non-linear models such as GATs, showcasing the robustness of graph prompting across different architectures. Our extensive experiments validate these theoretical results and confirm their practical implications, demonstrating that graph prompts can effectively leverage pre-trained models in various settings. By providing solid theoretical foundations, our work not only explains why graph prompts work but also guides the design of more effective prompting techniques. This empowers researchers and practitioners to utilize graph prompts with greater confidence, potentially leading to more efficient and generalized graph neural network models across diverse applications.

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

# APPENDIX

The appendix of this paper is organized as follows: Appendix A presents the detailed theoretical content on the main findings in the paper. In order to reduce the bar of reading this content, we first give some preliminaries and definitions in Appendix A.1, followed by fundamental lemmas (Appendix A.2) that will be used to prove our theorems. In Appendix A.3, we carefully prove the theorems in the main body of this paper, followed by a further mathematical discussion in Appendix A.4. Beyond theoretical analysis, Appendix B presents additional experimental results on the real-world datasets, which have similar observations to the main experiments in the paper. Appendix C introduces more details on the settings of the experiment.

## A   THEORETICAL ANALYSIS AND PROOFS

**Reading Guideline:** Appendix A "Theoretical Results and Proofs" is divided into four subsections:

- **A.1 Definitions and Preliminaries:** Readers are advised to initially skip this subsection. It serves as a reference for unfamiliar terms encountered later in the text.

- **A.2 Fundamental Lemmas:** These properties are essential components for proving the main theorems. We will clearly express what each lemma demonstrates. Readers are recommended to refer to this subsection when encountering these lemmas while reading the theorem proofs.

- **A.3 Detailed Proofs of Main Theorems:** This subsection is recommended as the primary focus for readers. It contains the core ideas behind why prompts work, even though some lemmas may be required for complete understanding.

- **A.4 Additional Mathematical Lemmas:** This subsection includes purely mathematical lemmas encountered during the proofs in A.2 or A.3. These lemmas are not directly related to GNN models or prompts. Readers are advised to review this subsection after reading the previous content.

### A.1   PRELIMINARIES AND DEFINITIONS

#### A.1.1   PRELIMINARIES

**Preliminary 1** (GCN and GAT)**.**

**Graph Convolutional Networks (GCN)** Kipf & Welling (2016) perform convolution operations on graph-structured data by aggregating feature information from a node's neighbors. The recursive update rule for a GCN layer is:

$$\mathbf{H}^{(i+1)} = \sigma\left(\tilde{\mathbf{A}}\mathbf{H}^{(i)}\mathbf{W}^{(i)}\right), \tag{7}$$

where $\tilde{\mathbf{A}} = \mathbf{D}^{-1/2}\mathbf{A}\mathbf{D}^{-1/2}$ is the normalized adjacency matrix, $\mathbf{H}^{(i)}$ is the node feature matrix at layer $i$, $\mathbf{W}^{(i)}$ is the learnable weight matrix, and $\sigma$ is a nonlinear activation function.

**Graph Attention Networks (GAT)** Veličković et al. (2017) enhance GCNs by introducing attention mechanisms to weigh the importance of neighboring nodes during aggregation. In the simplest form of self-attention, the attention coefficient between node $j$ and node $k$ is based on the inner product of their feature vectors:

$$e_{jk} = \mathbf{X}_j^\top \mathbf{X}_k, \tag{8}$$

$$\alpha_{jk} = \frac{\exp(e_{jk})}{\sum_{m \in \mathcal{N}(j)} \exp(e_{jm})}, \tag{9}$$

$$\mathbf{H}_j^{(i+1)} = \sigma\left(\sum_{k \in \mathcal{N}(j)} \alpha_{jk}\mathbf{H}_k^{(i)}\mathbf{W}^{(i)}\right), \tag{10}$$

where $\mathcal{N}(j)$ denotes the neighbors of node $j$, $\alpha_{jk}$ is the normalized attention coefficient, and $\mathbf{W}^{(i)}$ is the learnable weight matrix at layer $i$. This mechanism allows the model to focus on the most relevant neighbors when updating node representations.

**Preliminary 2** (Pre-training and Fine-tuning vs Pre-training and Prompt)**.**

**Pre-training and Fine-tuning** Zhu et al. (2021)involves two stages: first, a model is pre-trained on a large dataset to learn general representations, and then it is fine-tuned on a specific downstream task.

Formally, let $F_{\theta^*}$ be the pre-trained model and $C(\cdot)$ be the optimal mapping function that maps the original graph to the embedding vector of the downstream task (i.e., can be parsed to yield correct results for $G_o ri$ in the downstream task)

Fine-tuning aims to adjust the model parameters from $\theta^*$ to $\theta^{\#}$ so that:

$$F_{\theta^* \to \theta^{\#}}(G_{\text{ori}}) \approx C(G_{\text{ori}}). \tag{11}$$

**Pre-training and Prompting** keeps the pre-trained model parameters fixed and instead modifies the input data to bridge the pre-training and downstream tasks. Specifically, it seeks a transformation from the original graph $G_{\text{ori}}$ to a prompt-enhanced graph $G_{\text{bri}}$ such that:

$$F_{\theta^*}(G_{\text{bri}}) = C(G_{\text{ori}}). \tag{12}$$

This approach leverages the frozen pre-trained model by adapting the input data through graph prompts, eliminating the need to fine-tune the model parameters.

**Preliminary 3** (GPF and All-in-One)**.**

**Graph Prompt Frameworks** introduce learnable modifications to input graphs to enhance the performance of frozen pre-trained GNNs on downstream tasks. Two primary frameworks are **prompt token vectors** like *GPF/GPF-Plus* Fang et al. (2022) and **prompt subgraph** like *All-in-One* Sun et al. (2023a).

**GPF (Graph Prompt Feature)** adds a prompt vector to each node's feature vectors. Let $\boldsymbol{\omega} = \{p\}$, $p \in \mathbb{R}^{F \times 1}$ be the learnable prompt vector, then the updated node features are:

$$[\mathbf{X}_\omega]_i = \mathbf{X}_i + p \tag{13}$$

The original graph $G = (\mathbf{X}, \mathbf{A})$ becomes the prompt-enhanced graph $G_\omega = (\mathbf{X}_\omega, \mathbf{A})$. The prompt vector $p$ is optimized to minimize the loss on the downstream task:

$$p^* = \arg\min_p \sum_{G \in \mathcal{G}} \mathcal{L}_{T_{\text{dow}}}\left(F_{\theta^*}(P_{\boldsymbol{\omega}}(G))\right) \tag{14}$$

**GPF-Plus** adds a combination of multiple prompt vectors to each node's features. Let $\boldsymbol{\omega} = \{p_1, \cdots, p_k, Q\}$, $p_i \in \mathbb{R}^{F \times 1}$ be the learnable prompt vector, $Q \in \mathbb{R}^{M \times k}$. Let $P = \begin{pmatrix} p_1^\top \\ \vdots \\ p_k^\top \end{pmatrix}$.

The node features are updated in such a way:

$$[\mathbf{X}_\omega]_i = \mathbf{X}_i + Q_i P \tag{15}$$

The original graph $G_i = (\mathbf{X}, \mathbf{A})$ becomes the prompt-enhanced graph $G_{i,\omega} = (\mathbf{X}_\omega, \mathbf{A})$.

**All-in-One** incorporates entire prompt subgraphs into the original graph. Let $P \in \mathbb{R}^{k \times F}$ represent $K$ learnable prompt token vectors, and $\mathbf{A}_{\text{in}} \in \{0, 1\}^{k \times k}$ denote the internal adjacency among prompt tokens. The connections between prompt tokens and original nodes are defined by a cross adjacency matrix $\mathbf{A}_{\text{cro}} \in \{0, 1\}^{k \times N}$. The prompt-enhanced graph is:

$$G_\omega = (\mathbf{A} \cup \mathbf{A}_{\text{in}} \cup \mathbf{A}_{\text{cro}}, \mathbf{X} \cup \Omega) . \tag{16}$$

All-in-One optimizes the prompt tokens and their connections to adapt the pre-trained model to downstream tasks without altering the model parameters.

**Preliminary 4** (Diffusion Matrix)**.**

**Diffusion Matrix** Gasteiger et al. (2019) plays a crucial role in representing the diffusion process on a graph. Specifically, many GNN architectures can be expressed using the following formulation:

$$\mathbf{H} = \mathbf{S} \cdot \mathbf{X} \cdot \mathbf{W}, \tag{17}$$

Where: $\mathbf{S}$ is the diffusion matrix, derived from the graph's adjacency matrix and model structure, which governs how information propagates across the graph. $\mathbf{X}$ is the original node feature matrix. $\mathbf{W}$ is the learnable weight matrix associated with the GNN layer. $\mathbf{H}$ is the node feature embedding matrix after message transformation and aggregation.

### A.1.2 DEFINITIONS

We provide a glossary and default symbols meanings here for the reader's convenience.

| Term | Explanation |
|---|---|
| Bridge Set $B_G$ | $B_G = \{G_p \mid F_{\theta^*}(G_p) = C(G)\}$ |
| $\epsilon$-extended Bridge Set $\epsilon$-$B_G$ | $\epsilon$-$B_G = \{G_p \mid \|F_{\theta^*}(G_p) - C(G)\| \leq \epsilon\}$ |
| Adjacency matrix | A square matrix used to represent a finite graph, where $A_{ij} = 1$ if there is an edge from vertex $i$ to vertex $j$, and 0 otherwise |
| Diffusion matrix | The matrix equivalent to graph aggregation in GNNs |
| Span | For a set $V$, we say "p spans V" if p can take any value in V |
| $i$-th order embedding matrix | The embedding matrix after $i$ iterations of message passing and aggregation in a GNN |
| Cone | A set $C$ such that for any $x \in C$ and $\alpha \geq 0$, $\alpha x \in C$ |
| Convex set | A set $S$ such that for any $x, y \in S$ and $\alpha \in [0, 1]$, $\alpha x + (1 - \alpha)y \in S$ |
| Convex hull | The smallest convex set containing a given set of points |
| Graph Embedding Residual Vector | A vector representing the additional error in graph fitting due to non-linear components. For details, refer to the related A.3.5 |

Table 2: Glossary

| Symbol | Description |
|---|---|
| $G_{ori}$ | The original graph without prompting |
| $P_\omega$ | Graph prompting method with parameter $\omega$ |
| $C(G_{ori})$ | The optimal embedding vector for the downstream task. $C(\cdot)$ can be understood as the optimal downstream task model |
| $G_{bri}$ | The bridge graph that can be used to obtain $C(G_{ori})$ using the original model |
| $G_p$ | The graph after prompting |
| $G_\omega$ | The graph after prompting with parameter $\omega$ |
| $\epsilon$ | Represents the extent of Bridge set expansion, i.e. the "error" |
| $n$ | Generally represents the number of layers in the GNN |
| $F$ | Represents the dimension of the graph feature vector |
| $N$ | Represents the number of nodes in the graph |
| $M$ | Represents the number of graphs in the dataset $\Omega$ |
| $\Phi$ | Represents the aperture of the convex cone can be understood as the maximum opening angle |

Table 3: Symbol Table

### A.2 FUNDAMENTAL LEMMAS

This lemma serves as a foundational component for proving the theorem. However, readers may choose to skip the lemma initially and return to it when it is referenced in the theorem's proof.

**Default Case**  By default, we consider the GNN model as a surjective mapping operator from the graph set $\{G\}$ to $\mathbb{R}^F$, obtained after pre-training tasks. This operator can provide sufficient information to express the correct results of the graph in the pre-training task. (For instance, through a task-specific head mapping to the likelihood of a 0/1 decision.)

**Notations**  Here and in subsequent related content, we use $F_\theta$ to represent a GNN model whose aggregation process can be described by a diffusion matrix $S$. Here, $S$ is derived from the model type and the adjacency matrix $A$. For instance, in GCN, $S = A + \epsilon I$. The graph aggregation process is described as:

$$H^{(i)} = \sigma(S \cdot H^{(i-1)} \cdot W) \tag{18}$$

Where $H^{(0)}$ is exactly the node feature matrix $X$, and $H^{(i)}$ is referred to as the **i-th order embedding matrix**. Without loss of generality, we analyze the non-linear function $\sigma$ using Leaky ReLU. Other non-linear functions such as sigmoid can be analyzed similarly.

Denote the **transformation space** of $P_\omega$ and $G$ by $D_P(G) = \{P_\omega(G) | \omega \in \mathbb{R}^{|\omega|}\}$, where $\omega$ is the parameter of method $P$, $|\omega|$ denotes the dimensionality of the parameter $\omega$, and $P$ represents either GPF or All In One prompt method. We denote $P_\omega(G)$ as $G_\omega$, and the corresponding diffusion matrix and node feature matrix $S$ and $X$ after prompt as $S_\omega$ and $X_\omega$ respectively.

Hence, the **graph embedding vector** obtained from graph $G_\omega$ prompted by $P_\omega$ after passing through the GNN model $F_\theta$ can be denoted as $F_\theta(G_\omega)$.

Then, we can take each graph $G_\omega$ from the transformation space $D_P(G)$ and pass it through the GNN model $F_\theta$ to obtain the corresponding embedding vector. These embedding vectors can be collected into a set, the **transformation embedding vector set**:

$$\{F_\theta(G_\omega) | \text{ for all } G_\omega \in D_P(G)\} \tag{19}$$

Here, $F_\theta(G_\omega)$ can be expressed in formula form as:

$$F_\theta(G_\omega) = \text{ReadOut}(\sigma(S_\omega(\cdots \sigma(S_\omega X_\omega W) \cdots)W)) = \text{ReadOut}(H^{(n)}) \tag{20}$$

Where, the dots $(\cdots)$ indicate that the parentheses are nested $n$ times, representing $n$ iterations of the message passaging and aggregation. $n$ is the number of layers. $H^{(n)}$ is the n-th embedding matrix.

ReadOut process is viewed as the linear combination of the embedding matrix $H$, i.e. $\text{ReadOut}(H^{(n)}) = \mathbf{w}H^{(n)}$, where $\mathbf{w}$ is determined by the pre-trained model.

Specifically, we denote the process of obtaining the n-th order embedding matrix (i.e. final embedding matrix) as:

$$K_\theta(G_\omega) = H^{(n)} = \sigma(S_\omega(\cdots \sigma(S_\omega X_\omega W) \cdots)W) \tag{21}$$

### A.2.1  On the Range of Graph Embedding Matrix After Graph Prompting, one prompt node case

**Lemma 1** (Transformation of Graph after Prompt)**.**

Here we consider GPF and All-in-One methods. Without loss of generality, we assume that the prompt subgraph in All-in-One has only one node.

For GPF, the prompt vector is added to each node of graph $G$, while the topological connections of the graph remain unchanged. Therefore:

$$S_\omega = S, \quad X_\omega = X + \mathbf{1}_N \mathbf{p}^\top \tag{22}$$

Where $N$ is the number of nodes in the graph, $p$ is the prompt vector, $p \in \mathbb{R}^F$, $F$ represents the dimension of the parameter vector, $\mathbf{1}_N$ is a vector $\in \mathbb{R}^N$ with every component to be 1.

For All-in-One, the prompt subgraph is connected to graph $G$ in a parameterized way, i.e. there exist parameters that control the connection relationship between any two nodes in the prompt subgraph and the original graph.(as defined in A.1.1). Therefore:

$$S_\omega = \begin{pmatrix} S & l \\ l^\top & S_{NN} \end{pmatrix}, \quad X_\omega = X + \mathbf{e}_N \mathbf{p}^\top \tag{23}$$

Where $l \in \mathbb{R}^{N-1}$ is a column vector, $l_i \geq 0, \forall i \in \{1, \cdots, N-1\}$, $\mathbf{e}_N$ is a vector $\in \mathbb{R}^N$ with N-th component to be 1 and others to be 0. Kindly note that the number of graph nodes here is denoted as $N-1$ for consistency with GPF in form and $X \in \mathbb{R}^{N \times F}$ here is the natural extension of node attribute matrix $X_0 \in \mathbb{R}^{N-1 \times F}$ by adding an additional zero vector $\mathbf{0}_F \in \mathbb{R}^{1 \times F}$ like $\begin{bmatrix} X_0 \\ \mathbf{0}_F \end{bmatrix}$. In summary, the transformation of $X_\omega$ can be denoted as:

$$X_\omega = X + \mathbf{c}\mathbf{p}^\top, \quad \text{where } \mathbf{c}_i \geq 0 \tag{24}$$

Where  can be referred to as the coefficient vector.

**Lemma 2** (Range of Embedding Matrix after Nonlinear Transformation)**.**

Consider a weight matrix $W \in \mathbb{R}^{F \times F}$, and let $R(W)$ denote its row space. Suppose there exists a matrix $R$ such that each row of matrix $R$ is taken from the space $R(W)$. Let $\mathbf{c}$ be a vector with $c_i \geq 0$, and let $\mathcal{I}$ be the set of indices where $\mathbf{c}$ takes strictly positive values. Let $\mathbf{p}$ be vector spans $R(W)$(i.e. could take any value in $R(W)$).

Now, we consider $R + \mathbf{c}\mathbf{p}^\top$. **For** $i \in \mathcal{I}$, the $i$-th row of $R + \mathbf{c}\mathbf{p}^\top$ is: $R_i^\top + c_i\mathbf{p}^\top$ where $c_i > 0$, $R_i \in R(W)$, $\mathbf{p}$ spans $R(W)$. Therefore, $R_i^\top + c_i\mathbf{p}^\top \in R(W)$, and spans $R(W)$. **For** $i \notin \mathcal{I}$, $c = 0$, the $i$-th row of $R + \mathbf{c}\mathbf{p}^\top$ is simply $R_i^\top$.

Hence, $R + \mathbf{c}\mathbf{p}^\top$ can be written in the form of $R' + (\Delta R + \mathbf{c}\mathbf{p}^\top)$, where $\mathbf{p}$ spans $R(W)$, $R'$ represents the matrix with the rows whose index $i \notin \mathcal{I}$, and all other rows set to zero. $\Delta R = R - R'$.

Noted that every row of $\Delta R + \mathbf{c}\mathbf{p}^\top$ with index $i \in \mathcal{I}$, such row vector spans $R(W)$, as shown above, regardless of what the specific matrix $\Delta R$ is. Hence, the expressive power of graph prompting does not fundamentally differ for different $\Delta R$. We claim that from the perspective of embedding vectors (i.e., if the same embedding vectors can be produced through Readout, we don't distinguish the specific form of the matrix), we can simplify $\Delta R$ into $\mathbf{c}\mathbf{p}_0^\top$, where $\mathbf{c}$ is exactly the same $\mathbf{c}$ as the $\mathbf{c}$ in the assumption in the lemma, $\mathbf{p}_0 \in R(W)$ is a vector with the same size as $\mathbf{c}$. More detailed discussion refer to A.4. In this way, the calculation is greatly simplified.

Now, consider adding a nonlinear function $\sigma(\cdot) = $ Leaky-ReLU$(\cdot)$. We examine $\sigma(R' + \Delta R + \mathbf{c}\mathbf{p}^\top)$:

**Scenario 1:** When $W$ has full row rank, i.e., $\mathbf{p}^\top$, $\sigma(\mathbf{p}^\top)$ spans $\mathbb{R}^F$. In this case, for each row, we have:

For $i \in \mathcal{I}$

$$\sigma(R_i'^\top + \Delta R + c_i\mathbf{p}^\top) = \sigma(R_i'^\top + c_i\mathbf{p}^\top + c_i\mathbf{p}_0^\top) = \sigma(c_i(\mathbf{p}^\top + \mathbf{p_0}^\top))$$

since $R_i' = 0$. After the Leaky-ReLU transformation, $\sigma(c_i(\mathbf{p}^\top + \mathbf{p_0}^\top))$ can still span $\mathbb{R}^F$. Let's denote $\sigma(\mathbf{p}^\top + \mathbf{p_0}^\top)$ as $\mathbf{p}'^\top$, i.e. $\sigma(\mathbf{p}^\top + \mathbf{p_0}^\top) = \mathbf{p}'^\top$

For $i \notin \mathcal{I}$,

$$\sigma(R_i'^\top + c_i\mathbf{p}^\top + c_i\mathbf{p}_0^\top) = \sigma(R_i'^\top)$$

we use $\hat{R}'_i{}^\top$ to denote $\sigma(R_i'^\top)$ .

In conclusion, $\sigma(R' + \Delta R + \mathbf{c}\mathbf{p}^\top)$ can be written as $\hat{R}' + \mathbf{c}\mathbf{p}'^\top$, where $\mathbf{p}, \mathbf{p}'$ spans $\mathbb{R}^F$. Note that the property of $\mathbf{p}$ and $\mathbf{p}'$ is the same and there is a natural bijection between them as pointed out A.4. Without causing confusion in notation, we can use $\mathbf{p}$ to represent $\mathbf{p}'$ here.

**Scenario 2:** When $W$ is not full rank, $\mathbf{p}$ spans $R(W)$ space. In this case, row-wise, similar to Scenario 1:

$$\sigma(R' + \mathbf{c}\mathbf{p}^\top) = \hat{R}' + \mathbf{c}\mathbf{p}'^\top \tag{25}$$

where the set of all possible values of $\mathbf{p}'$ is $V_\alpha$, defined as:

$$V_\alpha = \{\text{Leaky-ReLU}(\mathbf{v}) \mid \mathbf{v} \in R(W)\} \tag{26}$$

**Lemma 3** (Range of Embedding Matrix after Prompt, Single Layer Case).

Consider a single-layer GNN model $F_\theta$. $K_\theta$ represents the embedding process for obtaining graph embedding matrix $H$. Then, we have:

$$
\begin{aligned}
K_\theta(P_\omega(G)) &= \sigma(S_\omega X_\omega W) \\
&= \sigma(S_\omega(X + \mathbf{c}\mathbf{p}^\top)W) \\
&= \sigma(S_\omega XW + S_\omega \mathbf{c}\mathbf{p}^\top W) \\
&= \sigma(R + \mathbf{c}'\mathbf{p}'^\top)
\end{aligned}
$$

Where $R = S_\omega XW$, each row of $R$ is a vector in the row space $R(W)$. $\mathbf{c}' = S_\omega \mathbf{c}$, $c'_i \geq 0$, $\mathbf{p}^\top W = \mathbf{p}'^\top$. Here, $\mathbf{p}'$ spans $R(W)$ since $\mathbf{p}$ spans $\mathbb{R}^N$. ($c'_i \geq 0$ since each element of $S_\omega$ is non-negative, the sum of each column in $S_\omega$ is strictly greater than 0 and the initial value of $\mathbf{c}$ is either $\mathbf{e}_N$ or $\mathbf{1}_N$. Note that the number of positive terms increases in $\mathbf{c}$.)

According to lemma 2, $\sigma(R + \mathbf{c}'\mathbf{p}'^\top)$ can be written as $\hat{R}' + \mathbf{c}'\mathbf{p}_\dagger^\top$. Here $\hat{R}'$ represents the element-wise Leaky-ReLu of the rows of $R$ with index $i \in \mathcal{I}$ and other rows take $\mathbf{0}.\mathbf{p}_\dagger^\top$ represents $\mathbf{p}^\top$ ($\mathbf{p}$ spans $\mathbb{R}^N$) if $W$ is of full row rank; $\mathbf{p}_\dagger^\top$ represents $\mathbf{p}'^\top$ ($\mathbf{p}'$ spans $V_\alpha$) if $W$ is non-full-rank, $V_\alpha = \{\text{Leaky-ReLU}(\mathbf{v}) \mid \mathbf{v} \in R(W)\}$.

Hence, in conclusion, we have: $K_\theta(P_\omega(G)) = \hat{R}' + \mathbf{c}'\mathbf{p}_\dagger^\top$

**Lemma 4** (Range of Embedding Matrix after Prompt, Multiple Layer Full Rank Case).

Consider a multiple-layer GNN model $F_\theta$. Then, we have:

$$
K_\theta(G_\omega) = \sigma(S_\omega(\cdots \sigma(S_\omega X_\omega W_1)\cdots)W_n) \tag{27}
$$

According to lemma 3, we have:

$$
\sigma(S_\omega X_\omega W_1) = \hat{R}' + \mathbf{c}'\mathbf{p}_\dagger^\top \tag{28}
$$

We are considering the full-rank case, i.e. $W$ is a full-rank matrix, according to lemma 3, we should take the equation for the full-rank case, which is:

$$
\sigma(S_\omega P_\omega W_1) = \hat{R}'_1 + \mathbf{c}_1\mathbf{p}^\top \tag{29}
$$

where $\mathbf{p}$ spans $\mathbb{R}^N$, $[\mathbf{c}_1]_i \geq 0$.

For this output, consider:

$$
\sigma(S_\omega(\hat{R}'_1 + \mathbf{c}_1\mathbf{p}^\top)W_2) = \sigma(S_\omega \hat{R}'_1 W_2 + \mathbf{c}_2\mathbf{p}^\top) \tag{30}
$$

where $\mathbf{c}_2 = S_\omega \mathbf{c}_1$, $[\mathbf{c}_2]_i > 0$. Compared with the equation in lemma 3, we find only $R$ has been replaced by $S_\omega \hat{R}'_1 W_2$, and components remain unchanged, so this lemma can be used again. Hence, we have:

$$
\sigma(S_\omega(\hat{R}'_1 + \mathbf{c}_1\mathbf{p}^\top)W_2) = \sigma(S_\omega \hat{R}'_1 W_2 + \mathbf{c}_2\mathbf{p}^\top) = \hat{R}'_2 + \mathbf{c}_2\mathbf{p}^\top \tag{31}
$$

where $\mathbf{p}$ spans $\mathbb{R}^N$.

Iteratively, we complete the entire $n$ aggregation processes of the GNN, and obtain:

$$
K_\theta(P_\omega(G)) = \hat{R}'_n + \mathbf{c}_n\mathbf{p}^\top \tag{32}
$$

where $\mathbf{p}$ spans the $\mathbb{R}^N$. (This formula can represent the expressiveness of prompt $P_\omega$ in the full-rank case)

**Remark 1.** *We can interpret this result as follows: (1) $\hat{R}'_n$ is related to the embedding matrix of the original graph $G$ (2) $\mathbf{c}_n \mathbf{p}^\top$ describes the additional range of the expression of GNN model after adding prompt.*

### A.2.2  ON THE RANGE OF GRAPH EMBEDDING MATRIX AFTER GRAPH PROMPTING, MULTIPLE PROMPT NODE CASE

**Lemma 5** (Range of Embedding Matrix after multiple prompt, Single Layer Case)**.**

We are considering single layer GNN $F_\theta$ here.

For the GPF or All in one Prompt Method, we can use the following uniform formula: we have $k$ independent prompt vectors $\mathbf{p}_i \in \mathbb{R}^F$ (or equivalently $k$ prompt nodes with $\mathbf{p}_i$ as its node feature), which form a $k \times F$ matrix $P$, where $P = \begin{pmatrix} \mathbf{p}_1^\top \\ \vdots \\ \mathbf{p}_k^\top \end{pmatrix}$. Denote by $M$ the number of graphs in the dataset $\Omega$. There exists an $M \times k$ coefficient matrix $Q$, each row vector of this coefficient matrix, denoted as $Q_i \in \mathbb{R}^k$, express how to linearly combine these $k$ vectors to add them to $i_{th}$ graph, i.e.:

$$G_{i,\omega} = (A_\omega, X + \mathbf{c} \cdot Q_i^\top P)$$

According to lemma 3, the embedding matrix for the $i_{th}$ graph is:

$$
\begin{aligned}
H_i &= K_\theta(G_{i,\omega}) \\
&= \sigma(S_{i,\omega} X_{i,\omega} W) \\
&= \sigma(S_{i,\omega}(X_{i,\omega} + \mathbf{1}_N Q_i^\top P)W) \\
&= \sigma(R + \mathbf{c}_{1,i} Q_i^\top P) \\
&= \sigma(R + \mathbf{c}_{1,i} \mathbf{p}_i^\top) \\
&= \hat{R}' + \mathbf{c}_{1,i} {\mathbf{p}'}_i^\top
\end{aligned}
$$

Where $\mathbf{p}'_i$ spans $\mathbb{R}^F$. (Implicit assumption is $W$ is full rank. we are discussing the upper limit of the expressive power of graph prompting, so we should use the full-rank model with stronger expressive power)

As discussed in lemma 2, we are considering from the perspective of embedding vectors. We claim that we can write $\mathbf{p}'_i = (Q'_i)^\top P'$, where ${Q'}_i^\top$ denotes the $i_{th}$ row of $Q'$, $Q'$ is a linear combination coefficient matrix, which is a mapping of $Q$. $P'$ spans $\mathbb{R}^{k \times F}$, which is a mapping of $P$. More detailed discussion is referred to A.4.

Without causing confusion in notation, we can directly use $P$, $Q$ to denote $P'$, $Q'$ here.

In summary, for a single-layer GNN with multiple prompts, the embedding matrix takes the form:

$$H_i = K(G_{i,\omega}) = \hat{R}' + \mathbf{c}_{1,i} \mathbf{p}_i^\top$$

where for any $i \in \{1, \ldots, M\}$, $\mathbf{p}_i = Q_i P$, and $\hat{R}'$ is defined consistently with Lemma 3.

**Lemma 6** (Range of Embedding Matrix after Prompt, Multiple Layer Case)**.**

We now consider a multiple-layer GNN model. For this model, we have:

$$K_\theta(G_{i,\omega}) = H_i^{(n)} = \sigma(S_{i,\omega}(\cdots \sigma(S_{i,\omega} X_{i,\omega} W) \cdots )W)$$

where, according to lemma 5,

$$\sigma(S_{i,\omega} X_{i,\omega} W) = \sigma(S_{i,\omega}(X_i + \mathbf{1}_N Q_i P)W) = \hat{R}' + \mathbf{c}_{1,i} p^\top$$

with $\mathbf{p}^\top = Q_i P$, and $P$ is the collection of $k$ prompts, spans $\mathbb{R}^{k \times F}$ space.

This form is consistent with lemma 5, except $R$ is replaced by $\hat{R}'$ and $\mathbf{c}_0$ is replaced by $\mathbf{c}_{1,i}$. Applying this result again, we get:

$$\sigma(S_{i,\omega}\sigma(S_{i,\omega}X_{i,\omega}W_1)W_2) = \sigma(S_{i,\omega}(\hat{R}' + \mathbf{c}_{1,i}Q_iP)W_2)$$
$$= \hat{R}'' + \mathbf{c}_{2,i}Q_iP$$

Where $P$ spans $\mathbb{R}^{k \times F}$.

Iterativly applying lemma 5 $n$ times, we obtain:

$$K_\theta(G_{i,\omega}) = H_i^{(n)} = \hat{R}^{(n)} + \mathbf{c}_{n,i}p_i^\top = \hat{R}^{(n)} + \mathbf{c}_{n,i}Q_i^\top P$$

where $P$ spans $\mathbb{R}^{k \times F}$, and $Q$ spans $\mathbb{R}^{M \times k}$.

In summary, for a multi-layer GNN with full-rank $W$ matrices in each layer, and for a dataset $\Omega$ with multiple graphs and use multiple prompts $P$, we have:

$$K_\theta(G_{i,\omega}) = \hat{R}^{(n)} + \mathbf{c}_{n,i}p_i^\top = \hat{R}^{(n)} + \mathbf{c}_{n,i}Q_i^\top P$$

where $P$ spans $\mathbb{R}^{k \times F}$, $Q$ spans $\mathbb{R}^{M \times k}$, for any $i \in \{1, \ldots, M\}$.

### A.2.3 ON THE RANGE OF GRAPH EMBEDDING MATRIX AFTER GRAPH PROMPTING, NOT FULL-RANK CASE

**Lemma 7** (Range of Embedding after Prompt, Multiple Layer Non-Full Rank Case).

We consider a multiple-layer GNN model $F_\theta$ with a potential non-full-rank weight matrix. We have:

$$K_\theta(G_\omega) = \sigma(S_\omega(\cdots \sigma(S_\omega X_\omega W) \cdots)W) \tag{33}$$

According to lemma 3, we have:

$$\sigma(S_\omega X_\omega W) = \hat{R}' + \mathbf{c}'\mathbf{p}_\dagger^\top \tag{34}$$

Since we are considering the non-full rank case, according to lemma 3, we should consider the non-full rank equation:

$$\sigma(S_\omega P_\omega W) = \hat{R}_1' + \mathbf{c}_1\mathbf{p}^\top \tag{35}$$

where $\mathbf{p}$ spans the set $V_\alpha^1$ as defined in lemma 3.

At this point, we can consider $\hat{R}_1'$ equal to zero. Since, for GPF, $\mathbf{c}_0 = \mathbf{1}_N$, and the monotonicity of $\mathbf{c}$ implies that each component of the vector $\mathbf{c}$ is strictly greater than 0. For All-in-One, $\mathbf{c} = \mathbf{e}_N$. We have $S_\omega = [s_1, \ldots, s_N]$, where $s_N = \begin{pmatrix} l \\ S_{NN} \end{pmatrix}$, $\mathbf{c}' = s_N$. we assume $l > 0$ component-wise(This can be achieved by adjusting parameters in All in one), then $\mathbf{c}'$ is component-wise $> 0$.

Recall that $R_1'$ only preserves rows where the corresponding component of $\mathbf{c} = 0$. Hence, $\hat{R}_1'$ can be considered to be zero.

For this output, we consider:

$$\sigma(S_\omega(\hat{R}_1' + \mathbf{c}_1\mathbf{p}^\top)W) = \sigma(S_\omega(\mathbf{c}_1\mathbf{p}^\top)W) = \sigma(\mathbf{c}_2\mathbf{p'}^\top) = \mathbf{c}_2\mathbf{p''}^\top \tag{36}$$

Here, $\mathbf{p'}^\top = \mathbf{p}^\top W$. This demonstrates that: (1). $\mathbf{p'}$ spans $V_\alpha^1 W$ (2). $\mathbf{p'} \in R(W)$. Hence, the set

of the range of $\mathbf{p}'$ is only a linear transformation to $V_\alpha^1$, i.e. $V_\alpha^1 W$. The set of the range of $\mathbf{p}''$ is operating Leaky-ReLU to the set $V_\alpha^1 W$, i.e. $\sigma(V_\alpha^1 W)$, denoted by $V_\alpha^2$.

Iteratively, We have:

$$F_\theta(G_\omega) = \mathbf{c}_n \mathbf{p}^{(n)\top} \tag{37}$$

where $\mathbf{p}^{(n)}$ spans $V_\alpha^n$. Specifically, according to lemma 12, $V_\alpha^n$ is a cone surface, i.e., for any $\mathbf{v} \in V_\alpha^n$ and $\lambda \in \mathbb{R}^+$, $\lambda \mathbf{v} \in V_\alpha^n$.

**Remark 2.** *Choosing $\hat{R}_1'$ to be zero is to simplify the calculation without affecting the theorem. Based on the principle that the minimum of the whole is less than or equal to the minimum of a part, under this assumption, we have obtained that the upper bound of the error for the prompt satisfy our assumption is certainly the upper bound for the optimal prompt.*

**Lemma 8** (Embedding Matrix Property, Multiple Layer GNN Non-Full Rank Case)**.**

We consider the GNN model $F_\theta$ be multi-layered and could have not a full-rank weight matrix. Then, the embedding matrix of graph $G$, according to the iterative formula of GNN, is:

$$K_\theta(G_\omega) = \sigma(S(\cdots \sigma(S \cdot X \cdot W) \cdots)W) \tag{38}$$

Let $S \cdot X$ be denoted as $X'$. Then $X'W$ is an $N \times F$ embedding matrix where each row is in the row space of $W$. According to the properties of Leaky-ReLU and the definition of $V_\alpha^1$, $\sigma(X'W) = Y'$ where each row vector is in the set $V_\alpha^1$.

Then $SY' = H'$, where each row of $H'$ is a positive linear combination of rows in $Y'$. Hence, each row of $H'$ should also be in the convex hull of cone surface $V_\alpha^1$. We name such a convex hull to be $C_\alpha^1$, and further, $C_\alpha^1$ is a convex cone. Based on lemma12, iteratively, we have each row of $H^{(i)}$ should also be in the convex hull $C_\alpha^i$ of cone surface $V_\alpha^i$.

We conclude the following: For any graph $G$, each row vector of the embedding matrix $H^{(n)}$, obtained after applying the GNN model $F_\theta$ to $G$, lies within a specific convex cone, whose surface is exactly $V_\alpha^n$ mentioned in lemma 7

**Remark 3.** *Lemma 8 characterizes the properties of each row vector in the embedding matrix after it has been processed by a non-full rank GNN model.*

## A.3 Proof of Theorem in the Paper

### A.3.1 Bridge Graph Existence Theorem

Proof of Theorem 1

*Proof.* For a given $G_{ori}$ and a downstream task $T_{dow}$, the embedding vector corresponding to the downstream task is formally defined as the embedding vector produced by the optimal downstream model for $T_{dow}$, which is thus uniquely determined.

Given our previous definition for the default case A.2, the $F_\theta$ discussed here is a surjective mapping from the graph space $\{G\}$ to $\mathbb{R}^F$. According to the properties of surjective mappings, for this particular $C(G_{\text{ori}}) \in \mathbb{R}^F$, there must exist a special graph $\hat{G}_{\text{bri}}$ such that:

$$F_\theta(\hat{G}_{\text{bri}}) = C(G_{\text{ori}}) \tag{39}$$

Upon examining the definition of $G_{\text{bri}}$, we find that $\hat{G}_{\text{bri}} = G_{\text{bri}}$. Theorem 1 is thereby proved. $\qquad \square$

### A.3.2 On Error-free Mapping to Bridge Set

Proof of Theorem 3 and 4

*Proof.* We aim to prove that there exists an optimal parameter $\omega_0$ such that the transformed graph obtained through the graph prompting method $P_{\omega_0}(G) \in B_G$. This is equivalent to proving that there exists $\omega_0$ such that $F_\theta(P_{\omega_0}(G_{ori})) = F_\theta(G_{bri}) = C(G_{ori})$, where the existence of $G_{bri}$ is guaranteed by Theorem 1.

Consider a multi-layer GNN model $F_\theta$ where each layer has full-rank matrices and non-linear function $\sigma(\cdot)$. According to lemma 4, regardless GPF or All-in-One prompt method, we have:

$$K_\theta(P_\omega(G_{ori})) = H^{(n)} = R^{(n)} + \mathbf{c}\mathbf{p}^\top$$

Where $c_i \geq 0$, $\|\mathbf{c}\| > 0$, $\mathbf{p}$ spans $\mathbb{R}^F$, $K_\theta(\cdot)$ denote the process of obtaining the embedding matrix, $R^{(n)}$ a matrix $\in \mathbb{R}^{N \times F}$, which is related to the embedding matrix of the original graph $G$ without prompting, $\omega$ denotes the parameter of the Prompt method.

Then, $F_\theta(P_\omega(G_{ori})) = \text{Readout}(H^{(n)})$. Considering the readout function as linearly combines the embedding vectors of nodes 1 to $n$ with certain weights: $\mathbf{w} = [w_1, \ldots, w_n]$, where $w_i > 0$, as defined in A.2, we have:

$$F_\theta(P_\omega(G_{ori})) = \sum_i w_i H_i^{(n)}$$

Then:

$$F_\theta(P_\omega(G_{ori})) = \mathbf{w}R^{(n)} + \mathbf{w}^\top \mathbf{c}\mathbf{p}^\top = R_0^\top + \lambda\mathbf{p}^\top$$

where $R_0^\top = \mathbf{w}R^{(n)}$, $\lambda = \mathbf{w}^\top \mathbf{c}$ ($\lambda > 0$), since $\mathbf{p}$ spans $\mathbb{R}^F$, we know that $F_\theta(P_\omega(G_{ori})) = R_0^\top + \lambda\mathbf{p}^\top$ is a surjective mapping from the range of $\omega$ to $\mathbb{R}^F$. Meanwhile, $F_\theta(G_{bri})$ is a fixed vector in $\mathbb{R}^F$.

By the surjective property, there must exist an $\omega_0$ such that:

$$F_\theta(P_{\omega_0}(G_{ori})) = F_\theta(G_{bridge}) = C(G)$$

This completes the proof. $\square$

### A.3.3 ON ERROR UPPER BOUND ANALYSIS OF MAPPING TO BRIDGE SET AT SINGLE GRAPH LEVEL

Proof of Theorems 5

*Proof.* (Prompt Error Upper Bound on Single Graph With GCN model with layers containing non-linear functions and possibly non-full rank matrices)

Consider the optimal $\omega_0$ such that the transformed graph $P_{\omega_0}(G) \in \epsilon\text{-}B_G$, where the "error" $\epsilon$ takes the minimum possible value: $\epsilon_0$.
We aim to prove Theorems 5 by showing that for any graph $G$ there exists an $\delta_0$ such that:

$$\frac{\|F_\theta(P_{\omega_0}(G_{ori})) - F_\theta(G_{bri})\|}{\|F_\theta(G_{bri})\|} < \delta_0$$

According to lemma 7 and 8, for both GPF and All-in-One prompts, there exists a convex cone $C$ such that:

$$K_\theta(G_\omega) = H^{(n)} = \mathbf{c}\mathbf{p}^\top$$

where $\mathbf{p}$ is on the surface of such convex cone, i.e., $\partial C$, and $F_\theta(G_{bri})$ is a vector inside the convex cone $C$.

Using the same readout method as in A.3.2, we have:

$$F_\theta(P_\omega(G_{ori})) = \sum_i w_i H_i^{(n)} = \mathbf{w}^\top \mathbf{c}\mathbf{p}^\top = \lambda \mathbf{p}^\top$$

where $\mathbf{p}$ spans surface of the convex cone ($\partial C$). Based on the properties of the cone surface, $\lambda \mathbf{p}$ is on the cone surface for any $\lambda > 0$, so we have $F_\theta(P_\omega(G_{ori}))$ spans the cone surface $\partial C$.

Consider $\|F_\theta(G_{bri}) - F_\theta(P_\omega(G_{ori}))\|$. The minimum value of this distance, according to definition 3, is the distance from an element in the convex cone to the cone surface, which is $\sin(\theta)\|v\|$, where $\theta$ represents the angle of $v$ to the surface.

This proves that $\frac{\|F_\theta(P_\omega(G_{ori})) - F_\theta(G_{bridge})\|}{\|F_\theta(G_{bridge})\|}$ has a upper bound, $\sin(\theta)$, where $\theta$ is related to both $v$ and $C$.

According to definition 4, we know that $\theta \leq \Phi/2$, where $\Phi$ represents the aperture of the cone $C$.

In summary:

$$\|F_\theta(P_\omega(G_{ori})) - C(G_{ori})\| = \|F_\theta(P_\omega(G_{ori})) - F_\theta(G_{bri})\| \tag{40}$$
$$\leq \sin(\Phi/2) \cdot \|F_\theta(G_{bri})\| \tag{41}$$
$$= \sin(\Phi/2) \cdot \|C(G_{ori})\| \tag{42}$$

The left part of the error upper bound ($\sin(\Phi/2)$) is only related to the model, while the right part is only related to the data ($G_{ori}$). This indicates that the error upper bound grows linearly with $\|C(G_{ori})\|$ by a coefficient, i.e. $\sin(\Phi/2)$.

Since the features magnitude and number of node of the graph data in the dataset have upper bounds, $\|C(G_{ori})\|$ also has an upper bound $C$. Hence, for the dataset, there is a uniform upper bound, which demonstrates that the model is effective and can utilize the powerful pre-trained model within a certain error range. □

### A.3.4  ERROR BOUND ANALYSIS OF MAPPING TO BRIDGE SET AT MULTIPLE GRAPH LEVEL

Proof of Theorems 6

*Proof.* We prove this theorem by considering the sum of squared of $\epsilon$, where $\epsilon$ denotes the degree of the bridge set extension. For simplicity, we will refer to it as the "error".

Consider a dataset with M graphs. We use either a single prompt vector GPF or a single node-prompted subgraph All-in-one approach. According to lemma 4, for each graph, we have the following formula:

$$K_\theta(G_i, \omega) = H_i^{(n)} = R_i^{(n)} + \mathbf{c}_i\mathbf{p}^\top \tag{43}$$

where $\mathbf{c}_i$ have the subscript $i$ to indicate that different graphs produce different $\mathbf{c}$, $\mathbf{p}$ spans $\mathbb{R}^F$. Note that $\mathbf{p}$ does not have a subscript $i$ because single Prompt Vector is used for all graphs, despite different graphs having different diffusion matrices $S$ and feature matrices $X$.

After performing the ReadOut operation, we get:

$$F_\theta(G_i) = \mathbf{w}^\top H_i^{(n)} = \mathbf{w}^\top R_i^{(n)} + \mathbf{w}^\top \mathbf{c}_i\mathbf{p}^\top = R_0^\top + \lambda_i\mathbf{p}^\top \tag{44}$$

Where $\lambda_i = \mathbf{w}^\top \mathbf{c}_i$. We can assume $R_0^\top$ is a zero matrix without loss of generality as discussed in lemma 7.

Now, consider the sum of squared "errors":

$$\sum_{i=1}^{M} \|C(G_i) - \lambda_i \mathbf{p}\|^2 = \sum_{i=1}^{M} \lambda_i^2 \|(1/\lambda_i)C(G_i) - \mathbf{p}\|^2 \tag{45}$$

$$= \sum_{i=1}^{M} \lambda_i^2 \|C_{\lambda_i}(G_i) - \mathbf{p}\|^2 \tag{46}$$

The optimal $\omega_0$ corresponds to correspond to minimizing the following loss value.

$$J = \sum_{i=1}^{M} \lambda_i^2 \|C_{\lambda_i}(G_i) - \mathbf{p}\|^2 \tag{47}$$

Which is the weighted sum of squared distances from $\mathbf{p}$ to $M$ different points in the space, denoted as: $D((C_{\lambda_1}(G_1), \ldots, C_{\lambda_n}(G_n)), (\lambda_1, \ldots, \lambda_n))$

The optimal $p$ is the weighted centroid of these $n$ vectors $(C_{\lambda_i}(G_i))_{i=1}^{M}$, as pointed out at Boyd & Vandenberghe (2004), i.e.:

$$\mathbf{p}^* = \frac{\sum_{i=1}^{M} \lambda_i C(G_i)}{\sum_{i=1}^{M} \lambda_i^2} \tag{48}$$

The closed-form expression for $J_{min}$ is:

$$J_{min} = \sum_{i=1}^{M} \|C(G_i) - \lambda_i \mathbf{p}^*\|^2 \tag{49}$$

Therefore, the minimum root mean squared of $\epsilon$ value("RMSE") $I(G_1, \ldots, G_n)$ satisfies

$$I(G_1, \ldots, G_M) = \min_\omega (\sqrt{\sum_{i=1}^{M} \lambda_i^2 \|C_{\lambda_i}(G_i) - \mathbf{p}\|^2 / M} = \sqrt{\sum_{i=1}^{M} \|C(G_i) - \lambda_i \mathbf{p}^*\|^2 / M} \tag{50}$$

$$= \sqrt{\frac{J_{min}}{M}} \tag{51}$$

Here, $I(G_1, \ldots, G_M)$ is a lower bound that is independent of the value of $\omega$, but is related to the distances between $C(G_i)$. This proves that the capability of a single prompt has an upper limit in this case, which proves Theorem 6. $\square$

Proof of Theorem 7

*Proof.* We validate this theorem by minimizing the sum of squared of $\epsilon$, where $\epsilon$ denotes the degree of the bridge set extension. For simplicity, we will refer to it as the "error".
Consider a dataset with M graphs. We use k prompt vectors for GPF or k node-prompted subgraphs. According to lemma 6, for each graph, we have:

$$H_i^{(n)} = K_\theta(G_i, \omega) = R_i^{(n)} + \mathbf{c}_i Q_i^\top P \tag{52}$$

where $P$ spans $\mathbb{R}^{k \times F}$, $Q$ represents the combination coefficients of different Prompts, and spans $\mathbb{R}^{M \times k}$. As discussed in Proof for Theorem 6, we can assume $R_i^{(n)}$ is a zero matrix for simplicity.

After the ReadOut operation:

$$F_\theta(G_i) = \mathbf{w}^\top H_i^{(n)} = \mathbf{w}^\top \mathbf{c}_i Q_i^\top P = \lambda_i Q_i^\top P \tag{53}$$

Since $Q_i$ spans $\mathbb{R}^k$, $\lambda_i Q_i = Q'_i$ also spans $\mathbb{R}^k$ ($\lambda_i > 0$). Thus, $F_\theta(G_i)$ spans the row space $R(P)$, which is (at most) a $k$-dimensional vector space determined by $k$ independent prompt vectors. Consider:

$$\sum_{i=1}^{M} \|C^\top(G_i) - Q'^\top_i P\|^2 = \sum_{i=1}^{M} \|C(G_i) - \mathbf{p}'_i\|^2 \tag{54}$$

where $\mathbf{p}'_i$ spans the row space of $P$. Equation 54 is the sum of squared distances from n vectors to the vector space $P$. We want to minimize:

$$J = \sum_{i=1}^{M} \|C(G_i) - \mathbf{p}'_i\|^2 \tag{55}$$

Let $S = [C(G_1), \ldots, C(G_M)]$ and $V = S^\top S$. According to lemma 11, the minimum value of $J$ is the sum of the $(k+1)$-th to $M$-th largest eigenvalues of $V$:

$$J_{min} = \sum_{i=k+1}^{M} \lambda_i^V \tag{56}$$

Where $\lambda_i^V$ denotes the $i_{th}$ largest eigenvalue of symmetric matrix $V$. Therefore, there exists an optimal $\omega_0$ such that the mean squared epsilon(error) is:

$$\mathcal{L}(G_1, \cdots, G_M) = \sqrt{\frac{\sum_{i=k+1}^{M} \lambda_i^V}{M}} \tag{57}$$

This indicates that in the multiple prompt framework, the optimal mean squared $\epsilon$ has an upper bound $\mathcal{L}(G_1, \cdots, G_M)$, which proves Theorem 7.

Notably, if $k \geq n$, we can find a suitable $P$ such that $F_\theta(G_i)$ and $C(G_i)$ are error-free for any $i \in \{1, \cdots, M\}$. This can be seen as an extension of Theorems 3 and 4. $\square$

**Remark 4.** *As Johnstone (2001) pointed out, the eigenvalues of $V = C^\top C$ in datasets often exhibit a truncated eigenvalue distribution. The first $k_0$ eigenvalues explain most of the variance. Furthermore, Jolliffe & Cadima (2016) shows that in many datasets, eigenvalues may exhibit exponential decay. Hence, the sum of remaining eigenvalues decreases rapidly as $k$ increases. This could explain why moderate-scale prompts are sufficient to achieve good results on large graph datasets.*

### A.3.5    ERROR DISTRIBUTION ANALYSIS OF MAPPING TO BRIDGE SET

Proof of Theorem 8

*Proof.* Consider a GNN model $F_\theta$ where the $W$ matrix in the last layer is not full rank. Denote by $F - r$ the rank of the $W$ weight matrix in the last layer. Consider a GPF prompt or a single node prompt in an All-in-one framework.

According to lemma 4, we have:

$$H^{(n-1)} = R^{(n-1)} + \mathbf{c}\mathbf{p}^\top \tag{58}$$

where $H^{(n-1)}$ is the $(n-1)_{th}$ order embedding matrix, $\mathbf{p}$ spans $\mathbb{R}^F$. As discussed in lemma 7, for simplicity, we can assume $R^{(n-1)} = 0$. Then:

$$K_\theta(G_\omega) = H^n(G_\omega) = \sigma(S_\omega(R^{(n-1)} + \mathbf{cp}^\top)W)$$
$$= \sigma(S_\omega \mathbf{cp}^\top W) \tag{59}$$
$$= \sigma(\mathbf{c}'\mathbf{p'}^\top)$$

where $\mathbf{p}'$ spans the row space $R(W)$.

Consider $K_\theta(G_{bri}) = H^{(n)}(G_{bri}) = \begin{pmatrix} \mathbf{v}_1^\top \\ \vdots \\ \mathbf{v}_n^\top \end{pmatrix}$, where $\mathbf{v}_i \in \mathbb{R}^F$.

Then, $\Delta F_\theta(G_{ori}) = F_\theta(G_\omega) - C(G_{ori}) = \mathbf{w}^\top H^{(n)}(G_\omega) - \mathbf{w}^\top H^{(n)}(G_{bri}) = \mathbf{w}^\top (H^{(n)}(G_\omega) - H^{(n)}(G_{bri})) = \mathbf{w}^\top \Delta H$.

Here, $\Delta H_{ij} = \sigma(c_i p'_j) - v_{ij} = \sigma(c_i p'_j - \sigma^{-1}(v_{ij}))$.

Therefore, $[\Delta F_\theta(G)]_j = \sum_i w_i \sigma(c_i p'_j - \sigma^{-1}(v_{ij}))$. This is a piecewise linear function of $p'_j$, denoted as $g(p) = \alpha(p)p - \beta(p)$ for simplicity, where $\alpha$ and $\beta$ are the coefficients of this piecewise linear function when $x$ takes the value $p$.

When $\omega$ (the graph prompt coefficient) is optimal, due to the independence of $p^j$, all $[\Delta F_\theta(G)]^j$ should take the minimum absolute value. That is, $p^*$ minimizes $|g(p_j^*)|$, for any $j \in \{1, \cdots, F\}$.

At this point, $\|\Delta F_\theta(G)\| = \sqrt{\sum_{j=1}^F g(p_j^*)^2}$, where $g(p_j^*) = \sum_{i=1}^N w_i \sigma(c_i p_j^* - \sigma^{-1}(v_{ij})) = \alpha_j p_j^* - \beta_j$.

Let $p'^* = \begin{pmatrix} \alpha_1 & \cdots & 0 \\ \vdots & \ddots & \vdots \\ 0 & \cdots & \alpha_F \end{pmatrix} p^*$ and $\boldsymbol{\beta} = [\beta_1, \ldots, \beta_F]^\top$.

Then $p'^* \in R(W)$, so $\|\Delta F_\theta(G)\| = \|p'^* - \boldsymbol{\beta}\|$. This value represents the distance between $\boldsymbol{\beta}$ and $p'^*$. Since $p'^*$ already minimizes $\|Ap' - \boldsymbol{\beta}\|$ and $Ap' \in R(AW)$, $\forall p' \in R(W)$, where $A$ denotes $diag(\alpha_1, \cdots, \alpha_F)$, we can estimate $\|p^{*'} - \boldsymbol{\beta}\|$ using the distance from $\boldsymbol{\beta}$ to the space $R(AW)$.

Consider the projection matrix $P$ onto the space $R(AW)$. Then $\|\Delta F_\theta(G)\| = \|(I - P)\boldsymbol{\beta}\|$.

We make the following assumption: when $p_j = p_j^*$, $\beta_j$ follows an i.i.d. normal distribution.

$$\beta_j = \sum_{i=1}^N -(\alpha)^k w_i v_{ij}, \quad \text{where } k = \begin{cases} 0 & \text{if } \sigma(c_i p'_j - \sigma^{-1}(v_{ij})) > 0 \\ -1 & \text{otherwise} \end{cases} \tag{60}$$

Where $\alpha$ is the is the parameter of the Leaky-ReLU. According to the Central Limit Theorem and the independence of different components in $\boldsymbol{\beta}$, it is reasonable to assume that $\boldsymbol{\beta}$ follows an $n$-dimensional i.i.d. normal distribution with mean 0 and variance $c$ for some positive constant $c$ as discussed in A.3.5.

Since $P$ is an $(F - r)$-dimensional projection matrix, according to lemma 13, we know that $\|(I - P)\boldsymbol{\beta}\|$ follows a Chi distribution with $r$ degrees of freedom and scalar $c$, i.e. $c\chi_r$. $\qquad \square$

**Remark 5.** *Here, our equation 58 implicitly assumes that the first $N - 1$ layers of GNN are of full rank. However, even if the rank of the first $N - 1$ layers of GNN could be non-full-rank, we still have $\mathbf{p}' \in R(W)$, according to lemma 3. Therefore, we can still use this theorem for approximation.*

Normal Distribution Assumption

We assume that the vector $\boldsymbol{\beta}$, which we call the graph embedding residual vector, follows an $F$-dimensional i.i.d. normal distribution with mean $\mathbf{0}$ and variance $c$:

$$\boldsymbol{\beta} \sim \mathcal{N}(0, cI_F) \tag{61}$$

where $I_F$ is the $F \times F$ identity matrix.

Intuitively, this graph embedding residual vector $\boldsymbol{\beta}$ represents the additional term that the graph needs to fit due to non-full-rank and non-linear components. Formally, its components are defined as:

$$[\boldsymbol{\beta}]_j = \sum_{i=1}^{N} -(\alpha)^k w_i v_{ij}, \quad \text{where } k = \begin{cases} 0 & \text{if } \sigma(c_i p'_j - \sigma^{-1}(v_{ij})) > 0 \\ -1 & \text{otherwise} \end{cases} \tag{62}$$

The reasons behind this assumption are as follows: (1). Symmetry (2). The sum of random variables and Central Limit Theorem: Each $\beta_j$ is a sum of multiple terms, each of which can be considered a random variable. The Central Limit Theorem suggests that its distribution should approach a normal distribution. (3). Continuous and smooth distribution (4). Independence: The components of $\boldsymbol{\beta}$ are assumed to be independent due to the independence of the input features and the structure of the GNN.

Therefore, it is reasonable to assume that $\boldsymbol{\beta}$ follows an $F$-dimensional i.i.d. normal distribution with mean 0 and variance $c$ for some positive constant $c$.

### A.3.6 ANALYSIS OF NONLINEAR GRAPH NEURAL NETWORKS

Proof of Theorem 9

*Proof.* According to Preliminary A.1.1, we choose the simplest form of GAT here, with only a self-attention mechanism. We consider the diffusion matrix $S$ as a weighted adjacency matrix, where each entry $S_{ij}$ represents the coefficient of the edge between node $i$ and node $j$. Such coefficient is a non-negative scalar, which can be understood as the weight of the edge connecting node $i$ and node $j$, $S \in [0,1]^{N \times N}$. Here, single-node All-in-one connection matrices $A_{in} \in [0,1]^{1 \times 1}$ and $A_{cro} \in [0,1]^{N \times 1}$ respectively. Then, we have such an iterative formula:

$$H = \sigma((S_\omega \odot \langle X_\omega, X_\omega \rangle) \cdot X_\omega \cdot W)$$

where $\odot$ denotes the Hadamard product, representing element-wise multiplication of two matrices, $\langle \cdot, \cdot \rangle$ represents the inner product of 2 matrices. In this case, we have:

$$X_\omega = \begin{pmatrix} X \\ \mathbf{p}^\top \end{pmatrix}$$

$$\langle X_\omega, X_\omega \rangle = \begin{pmatrix} X^\top X & X\mathbf{p} \\ \mathbf{p}^\top X^\top & \mathbf{p}^\top \mathbf{p} \end{pmatrix}$$

$$S_\omega = \begin{pmatrix} S & l \\ l^\top & 0 \end{pmatrix}$$

Where $l$ denotes the cross adjacency matrix $A_{cro} \in [0,1]^N$. We set $A_{in}$ to 0, hence, $S_{NN}$ is 0. $\mathbf{p}$ is the prompted vector or node feature. Then, we have:

$$S_\omega \odot \langle X_\omega, X_\omega \rangle W = \begin{pmatrix} S \odot \langle X, X \rangle & l \odot (X \cdot \mathbf{p}) \\ l^T \odot (X \cdot \mathbf{p})^T & 0 \end{pmatrix} \cdot \begin{pmatrix} X \\ \mathbf{p}^\top \end{pmatrix} \cdot W \tag{63}$$

$$= \begin{pmatrix} S \odot \langle X, X \rangle & X \cdot (l \odot \mathbf{p}) \\ l^T \odot (X \cdot \mathbf{p})^T & 0 \end{pmatrix} \cdot \begin{pmatrix} X \\ \mathbf{p}^\top \end{pmatrix} \cdot W \tag{64}$$

For simplification, we consider only updating the embedding of rows of original nodes in the embedding matrix. Then we can obtain:

$$H[1:N,] = [\begin{pmatrix} S \odot \langle X, X \rangle & X \cdot (l \odot \mathbf{p}) \\ l^T \odot (X \cdot \mathbf{p})^T & (\mathbf{p}^\top \mathbf{p}) \end{pmatrix} \cdot \begin{pmatrix} X \\ \mathbf{p}^\top \end{pmatrix} \cdot W][1:N,] \tag{65}$$

$$= \sigma((S \odot \langle X, X \rangle \quad X(l \odot \mathbf{p})) \cdot \begin{pmatrix} X \\ \mathbf{p}^\top \end{pmatrix} \cdot W) \tag{66}$$

$$H[N+1] = \mathbf{p}^\top \tag{67}$$

Here, we can choose $l_i = \frac{c_i}{p_i}$. Then, we have:

$$H[1:N,] = \sigma((S \odot \langle X, X \rangle \quad \mathbf{c}) \cdot \begin{pmatrix} X \\ \mathbf{p}^\top \end{pmatrix} \cdot W) \tag{68}$$

$$= \sigma(S'XW + \mathbf{c}'\mathbf{p}^\top W) \tag{69}$$

$$= R + \mathbf{c}\mathbf{p}^\top \tag{70}$$

$$H[N+1,] = \mathbf{p}^\top \tag{71}$$

$$H = R' + \mathbf{c}\mathbf{p}^\top \tag{72}$$

$$\tag{73}$$

where $\mathbf{c}$ is chosen independent of $\mathbf{p}$, $\mathbf{c}' = \begin{pmatrix} \mathbf{c} \\ 1 \end{pmatrix}$. This translates to the iterative formula we have discussed earlier in 2. By analogy with the proofs of Theorems 3, 4, and 5, we can obtain similar Theorem 9. $\qquad\square$

### A.4 FURTHER MATHEMATICAL DISCUSSION

**Discussion on p:** In lemma 2, we rely on the claim that assumes $\Delta R = \mathbf{c}\mathbf{p}_0^\top$ won't lose the generality. This assumption is mainly for simplification, allowing us to obtain $\sigma(\Delta R + \mathbf{c}\mathbf{p}^\top) = \mathbf{c}\mathbf{p}'^\top$, $\mathbf{p}, \mathbf{p}'$ spans $R^F$.

The core purpose of this assumption is that when $\Delta R = \mathbf{c}\mathbf{p}^\top$, the $\tau_{ij}$ below is invariant to $i$.

$$[\sigma(R + cp^\top)]_{ij} = \begin{cases} [R + \mathbf{c}\mathbf{p}^\top]_{ij} & \text{when } p_j \geq \tau_{ij} \\ \alpha[R + \mathbf{c}\mathbf{p}^\top]_{ij} & \text{when } p_j < \tau_{ij} \end{cases}$$

Use $\tau_{ij}$ is independent of $i$, we can immediately establish a one-to-one mapping between $\mathbf{p}'$ and $\mathbf{p}$, (i.e., $p'_j = \begin{cases} p_j - \tau_j & \text{when } p_j \geq \tau_j \\ (p_j - \tau_j)/\alpha & \text{when } p_j < \tau_j \end{cases}$ ), thereby successfully establishing an equivalence between changes in $\mathbf{p}$ at the input end and the $\mathbf{p}'$ at the output end. This is what lemma 2 is doing. Noted that $\Delta R = \mathbf{c}\mathbf{p}_0^\top$ don't hold in general.

Now let us prove that the specific form of $R$ does not affect the results based on ReadOut. This is equivalent to proving the following lemma:

**Lemma 9** (Equivalence of R under ReadOut). *Considering a vector $\mathbf{p}$ spans $R^F$, then **(1)** ReadOut$(\sigma(R + \mathbf{c}\mathbf{p}^\top))$ spans $R^F$. **(2)** Given $\mathbf{p}_i$ is independent of each other, then $[ReadOut(\sigma(R + \mathbf{c}\mathbf{p}^\top))]_i$ is independent of each other.*

Consider the ReadOut of the Output Matrix $\sigma(R + \mathbf{c}\mathbf{p}_0^\top)$, i.e. ReadOut$(\sigma(R + \mathbf{c}\mathbf{p}^\top))$. Then the $j_{th}$ compoenet of such a $\mathbb{R}^F$ vector is:

$$\sum_{i=1}^N w_i[\sigma(R + \mathbf{c}\mathbf{p}^\top)]_{ij} = \sum_{i=1}^N [w_i(R_{ij}(p_j) + c_{ij}(p_j)p_j)]$$

where $R_{ij}(p) = \begin{cases} R_{ij} & \text{when } p \geq -R_{ij}/c_i \\ \alpha R_{ij} & \text{otherwise} \end{cases}$, $c_{ij}$ likewise.

Here, subscript $i$ indicates summing by row. Then, considering each column $j$, we have $F$ independent functions of $p_j$:

$$p'_j(p_j) = \sum_{i=1}^{N} w_i(R_{ij}(p_j) + c_{ij}(p_j)p_j) = R'_j(p_j) + C'_j(p_j)p_j$$

Where $p'_j(\cdot)$ is a piece-wise linear function of $p_j$ and $C'_i$ and $R'_i$ are corresponding piecewise linear coefficients of $p_j$. such a piecewise linear function takes values in $(-\infty, +\infty)$. Considering the independence of $F$ columns, it follows that **(1)** $\text{ReadOut}(\sigma(R + \mathbf{c}\mathbf{p}^\top))$ spans $\mathbb{R}^F$ **(2)** each component of $[\text{ReadOut}(\sigma(R + \mathbf{c}\mathbf{p}^\top))]$ is independent of each other. Which is what we need for the proofs of Theorems 3 and 4.

**Discussion on $Q$ and $P$:** As lemma 7 pointed out, we may use the same notations $Q$ and $P$ in the preceding and following layers, but they have different meanings. We write it to simplify the expressions. To verify it is valid under ReadOut perspective, as A.4 pointed out, the following lemma is required:

**Lemma 10.** *(Equivalence of $P$ and $Q$ under ReadOut) Consider a matrix $P \in \mathbb{R}^{k \times F}$, a matrix $Q \in \mathbb{R}^{M \times k}$, $\mathbf{p_i}^\top = Q_i^\top P$ then ReadOut$(\sigma(\mathbf{c}\mathbf{p_i}^\top))$ spans $\mathbb{R}^F$.*

According to lemma 9, we have:

$$[C(\mathbf{p_i})]_j = [\text{ReadOut}(\sigma(\mathbf{c}\mathbf{p_i}^\top))]_j = \sum_{i=1}^{N} w_i c_i \sigma([\mathbf{p_i}]_j) = \lambda \sigma([\mathbf{p_i}]_j)$$

Where $\mathbf{p_i} = Q_i P$, $w_i$ is the coefficient of ReadOut process, $C(\mathbf{p_i})$ denotes the embedding vector after ReadOut, $\lambda$ denotes $\sum_{i=1}^{N} w_i c_i$.

Hence, every component (with subscript $j$) of the embedding vector $C(\cdot)$ is a piece-wise linear function of $[\mathbf{p_i}]_j$. Specifically, this function is:

$$p'(p_j) = \begin{cases} \lambda p_j & \text{if } p_j \geq 0 \\ \alpha \lambda p_j & \text{otherwise} \end{cases}$$

where $\alpha$ is the coefficient of the leaky ReLU. range of such a function is $(-\infty, +\infty)$. Since $F$ components of $C(\mathbf{p_i})$ is independent, ReadOut$(\sigma(\mathbf{c}\mathbf{p_i}^\top)) = C(\mathbf{p_i})$ spans $\mathbb{R}^F$.

**Specific form of the bijective mapping of $P$ and $Q$**

Then, for each $i \in \{1, \cdots, M\}$, we have a Embedding Vector $C(\mathbf{p_i})$. Considering putting such Embedding Vectors into a matrix, i.e., $\tilde{C} = \begin{pmatrix} C(p_1) \\ \vdots \\ C(p_M) \end{pmatrix}$. Then:

$$\tilde{C} = \text{diag}(\lambda_1, \ldots, \lambda_n) \begin{pmatrix} \sigma(\mathbf{p_1}^\top) \\ \vdots \\ \sigma(\mathbf{p_M}^\top) \end{pmatrix} \tag{74}$$

$$= \text{diag}(\lambda_1, \ldots, \lambda_n) \begin{pmatrix} \alpha^{I_{11}}[\mathbf{p_1}]_1 \cdots \alpha^{I_{1F}}[\mathbf{p_1}]_F \\ \vdots \\ \alpha^{I_{M1}}[\mathbf{p_M}]_1 \cdots \alpha^{I_{MF}}[\mathbf{p_M}]_F \end{pmatrix} \tag{75}$$

$$\tag{76}$$

Here, we claim that we can take $I_{ij} = I_j$ by 6, i.e. for any $i \in \{1, \cdots, M\}$, $I_{ij} = I_j$.

Then we have:

$$\tilde{C} = \text{diag}(\lambda_1, \ldots, \lambda_n) \begin{pmatrix} [\mathbf{p_1}]_1 \cdots [\mathbf{p_1}]_F \\ \vdots \\ [\mathbf{p_M}]_1 \cdots [\mathbf{p_M}]_F \end{pmatrix} \text{diag}(\alpha^{I_1}, \cdots, \alpha^{I_F}) \tag{77}$$

$$= \text{diag}(\lambda_1, \ldots, \lambda_n) \cdot \begin{pmatrix} \mathbf{p_1}^\top \\ \vdots \\ \mathbf{p_M}^\top \end{pmatrix} \cdot \text{diag}(\alpha^{I_1}, \cdots, \alpha^{I_F}) \tag{78}$$

$$= \text{diag}(\lambda_1, \ldots, \lambda_n) \cdot QP \cdot \text{diag}(\alpha^{I_1}, \cdots, \alpha^{I_F}) \tag{79}$$

$$= Q'P' \tag{80}$$

Where $P'$ and $Q'$ give out the specific form of the bijective mapping between $P$ and $Q$ in lemma 5.

**Remark 6.** *We demonstrate that the claim in Explanation A.4 does not affect the results in Theorems 6 and 7.*

*Theorem 6 is about the case of a single prompt, making the claim trivially true.*

*For Theorem 7, we are seeking an upper bound for the optimal case. Among all prompt parameters satisfying the claim, if we can obtain the result in 7. Then, based on the principle that the global minimum is less than or equal to any partial minimum, the error in the optimal prompt would only be smaller, making such upper bound smaller.*

*In conclusion, our assumption is valid.*

**Lemma 11** (Minimum Sum of Squared Distances from M Vectors to a k-dimensional Subspace)**.**

Suppose we have M vectors in $\mathbb{R}^F$ and a k-dimensional subspace of $\mathbb{R}^F$. We consider the minimum sum of squared distances from these M vectors to the k-dimensional subspace. This is equivalent to minimizing the following objective:

$$J = \|X - XWW^T\|_F^2 = \text{Tr}((X - XWW^T)^T(X - XWW^T))$$

Where $W$ is a $k \times F$ matrix. According to Hotelling (1933), let: $X^T X = V\Lambda V^T$ where $V$ is the matrix of eigenvectors and $\Lambda$ is the diagonal matrix of eigenvalues. The optimal $W$ is then the first $k$ columns of $V$: $W = V[:, : k]$. Substituting this into the objective function:

$$J = \text{Tr}(V\Lambda V^T - V\Lambda V^T VV[:, : k]V[:, : k]^T$$
$$- VV[:, : k]V[:, : k]^T\Lambda V^T + VV[:, : k]V[:, : k]^T\Lambda V^T V[:, : k]V[:, : k]^T)$$

Simplifying, we get:

$$J = \text{Tr}(\Lambda - \Lambda[: k, : k] - \Lambda[: k, : k] + \Lambda[: k, : k])$$
$$= \text{Tr}(\Lambda) - \text{Tr}(\Lambda[: k, : k])$$
$$= \sum_{i=k+1}^{M} \lambda_i$$

Therefore, the minimum sum of squared distances from M vectors to a k-dimensional subspace is $\sum_{i=k+1}^{M} \lambda_i$, where $\lambda_i$ is the $i$-th eigenvalue of $X^T X$.

**Remark 7.** *This result is essentially the property of dimensionality reduction of M vectors to k-dimensional.*

**Definition 3** (Minimum Projection Distance to Cone Surface)**.**

Given a cone $C \subseteq \mathbb{R}^n$ with surface $\partial C$, and any vector $\mathbf{v} \in C$, we define $L(\mathbf{v}, \partial C)$ as the minimum distance (projection distance) from $\mathbf{v}$ to the surface of $C$. Formally:

$$L(\mathbf{v}, \partial C) = \min_{\mathbf{x} \in \partial C} \|\mathbf{v} - \mathbf{x}\| \tag{81}$$

**Definition 4** (The angle to boundary and the aperture of the cone).

For any interior point $x$ of a convex cone $C$, we define:

(i) The minimum distance from $x$ to the boundary of $C$:

$$d(x, \partial C) = \min\{\|x - y\| \mid y \in \partial C\} = d(x, \text{proj}_C(x)),$$

where $\text{proj}_C(x)$ is the projection of $x$ onto $C$.

(ii) The cosine of the angle $\theta$ between $x$ and the boundary of $C$:

$$\cos \theta_{x,C} = \frac{\langle x, \text{proj}_C(x) \rangle}{\|x\| \|\text{proj}_C(x)\|}.$$

(iii) The relationship between the distance and the angle $\theta$:

$$d(x, \partial C) = \|x\| \sin \theta_{x,C}.$$

(iv) The aperture $\Phi$ of the cone

$$\sin(\Phi) = \max_{v,u \in C}(|\frac{<v, u>}{\|v\| \cdot \|u\|}|)$$

Moreover, for any interior point $x$, we have $\theta_{x,C} \leq \Phi/2$. This formulation is based on the Distance-Support Theorem by Rockafellar (1970).

**Lemma 12** (Relationship between Range of Prompted and Embedding matrix).

As mentioned in lemma 3, $V_\alpha^1 = V_\alpha = \{\sigma(\mathbf{v}) \mid \mathbf{v} \in R(W)\}$. In lemma 8, under the alternating action of the non-linear function Leaky-ReLU and linear transformation $W$, we have $V_\alpha^i = \sigma(V_\alpha^{i-1}W)$.

**First** prove $V_\alpha^i$ is a cone surface. For $V_\alpha^1$, given $\lambda > 0$ and $\mathbf{v} \in R(W)$, we have $\sigma(\lambda \mathbf{v}) = \lambda \sigma(\mathbf{v})$. Thus, if $\sigma(\mathbf{v}) \in V_\alpha^1$, then $\forall \lambda > 0$, $\lambda \sigma(\mathbf{v}) \in V_\alpha^1$. Furthermore, consider $\mathbf{v} \in V_\alpha^{i-1}$, $\forall \lambda > 0$, $\sigma((\lambda \mathbf{v})W) = \lambda \sigma(\mathbf{v}W)$. It follows that, $\forall \mathbf{v} \in V_\alpha^i, \forall \lambda > 0$, $\lambda \mathbf{v} \in V_\alpha^i$, hence, $V_\alpha^i$ is a cone surface. Iteratively, we obtain that $V_\alpha^i$ is a cone surface for $i \in \{1, \ldots, n\}$.

**Next**, consider $C_\alpha^i$ as the convex hull of $V_\alpha^i$. Specifically, since $V_\alpha^i$ is a cone surface, $C_\alpha^i$ is a convex cone. By linearity of multiple a matrix, $C_\alpha^i W$ remains a convex cone. After applying $\sigma(\cdot)$, this convex hull transforms into another convex cone or cone surface, namely, $\sigma(C_\alpha^i W)$, and the corresponding $V_\alpha^{i+1}$ is the surface of $C_\alpha^{i+1}$. $\sigma(C_\alpha^i)$ is then the convex hull of $\sigma(C_\alpha^i W)$.

After $n$ iterations of the aggregation, we can conclude that: $V_\alpha^i$ is a cone surface, $C_\alpha^i$ is a convex cone, and $V_\alpha^i$ is the surface of $C_\alpha^i$.

**Remark 8.** *Here we assume that the process of obtaining the convex hull $C_\alpha^{i+1}$ from $\sigma(C_\alpha^i W)$ keeps the $V_\alpha^{i+1}$ to be the surface of $C_\alpha^{i+1}$. This always holds true unless the nonlinear transformation $\sigma(\cdot)$ turns the convex cone $V_\alpha^i W$ non-convex, which is unlikely when piece-wise linear function Leaky-ReLU is used as the nonlinear function as discussed in Ben-Tal & Nemirovski (2001).*

**Lemma 13** (Distribution of the Norm of an n-Dimensional i.i.d. Normal Vector After Multiplying Projection Matrix).

Let $P$ be a projection matrix onto a linear subspace $V$. Then $I - P$ is a projection matrix onto the orthogonal complement of $V$. Assume $\varsigma$ is an $n$-dimensional random vector following a normal distribution with mean $\mathbf{0}$ and covariance matrix $cI_F$, where $c$ is a positive constant. i.e., $\varsigma \sim \mathcal{N}(\mathbf{0}, cI_F$

Let $\zeta = (I - P) \cdot \varsigma$, according to the linear property of Gaussian Random vector, we have $\zeta \sim N(\mu', \Sigma')$, where:

$$\mu' = (I - P) \cdot \mu$$
$$= \mathbf{0}$$
$$\Sigma' = (I - P)^T \cdot \Sigma \cdot (I - P)$$
$$= c(I - P)^2 = c(I - P)$$

Denote by $r$ the rank of matrix $I - P$.

Apply decorrelation to $\zeta$, we have $\iota = U^T \cdot \zeta$, $\iota \sim N(\vec{0}, \Sigma'')$, where $\Sigma'' = \begin{pmatrix} c^2 \cdot I_r & O \\ O & O_{F-r \times F-r} \end{pmatrix}$, $U$ is a orthogonal matrix. Hence,

$$\|(I - P)\varsigma\| = \|\iota\| = \sqrt{\sum_{i=1}^{r} \iota_i^2}$$

Since $\sum_{i=1}^{r} \iota_i^2 \sim \chi^2(r)$, we have: $\|\zeta\| = \sqrt{\sum_{i=1}^{r} \iota_i^2} \sim \chi(r)$

It follows that the norm of an n-dimensional i.i.d. normal vector after multiplying projection matrix: $\|(I - P)\varsigma\|$ obey Chi-Distribution with r degrees of freedom.

## B SUPPLEMENTARY EXPERIMENTS

We conducted supplementary experiments on real-world datasets to validate our results. We performed similar experiments as in the main text using entirely different datasets, namely NCI1 and DD Morris et al. (2020). The NCI1 dataset has 37-dimensional node features, 4,110 subgraphs, an average of 29.87 nodes per subgraph, and an average of 64.60 edges per subgraph. The DD dataset has 89-dimensional node features, 1,178 subgraphs, an average of 284.32 nodes per subgraph, and an average of 1,431.32 edges per subgraph. Our results are shown in the following figures.

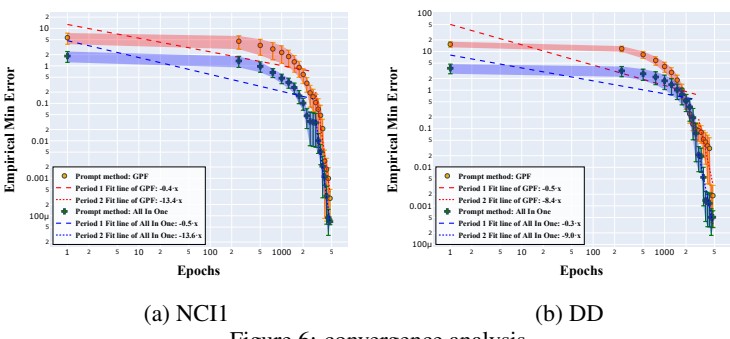

(a) NCI1           (b) DD

Figure 6: convergence analysis

**Exp1** we examined the loss curves during training for GPF and All-in-one prompting methods with full-rank matrices. Similar to observations from synthetic datasets, the prompt method shows a period of steady decline followed by a rapid decrease to a small magnitude, then slowly converging to zero. This aligns with our theoretical results.

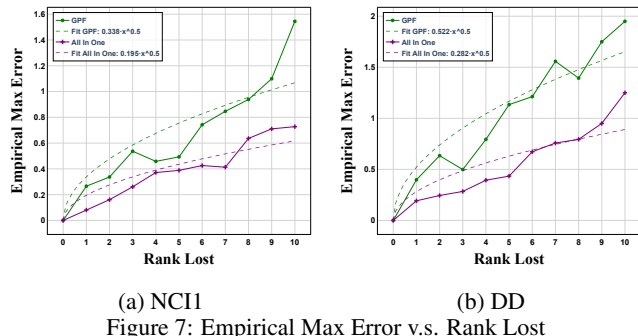

(a) NCI1           (b) DD

Figure 7: Empirical Max Error v.s. Rank Lost

**Exp 2** we investigated the relationship between **empirical maximum error** and rank loss for non-full-rank matrices. The results here are similar to those in the main text, with loss (empirical maximum error) increasing as rank loss increases. Where rank loss is $r$ means the rank of weight matrix $W$ in GNN is $n - r$.

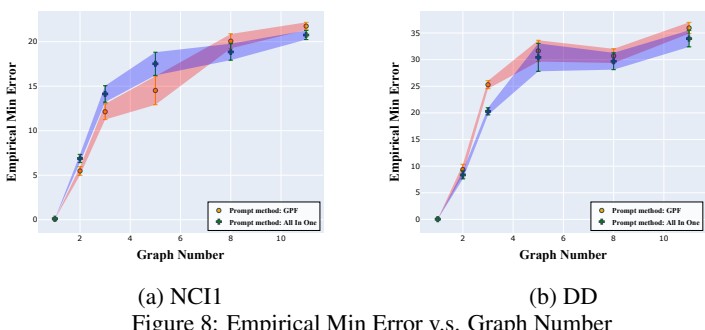

(a) NCI1           (b) DD

Figure 8: Empirical Min Error v.s. Graph Number

**Exp 3** We examined how the RMSE of GPF or *single-node* All-in-one Prompt methods change as the number of graphs increases in multiple graph mapping scenarios. We observed that in real-world datasets, RMSE also increases rapidly as the number of graphs increases, which aligns with our observation in the main experiment.

## C   EXPERIMENT DETAILS

**Data Preparation:** We first confirm our theoretical findings on synthetic datasets because these datasets offer controlled environments, allowing us to isolate specific variables and study their impacts. We generate these datasets by defining the dimension of graph feature vector($F$), average of graph node numbers ($N_{avg}$), average of graph edge numbers ($E_{avg}$), and number of graphs in the dataset ($M$). These parameters characterize both individual graphs and the entire dataset, facilitating our study of the relationship between these features and $\epsilon$. In detail, the distribution of graph node feature vectors is set to a normal distribution $\mathcal{N}(0, 1)$; the graph edge density $\rho$ is set to $0.15$, where $\rho$ represents the probability of an edge existing between any two nodes.

**Model Settings:** To evaluate our approach, we conduct experiments using two representative Graph Neural Network (GNN) architectures: Graph Convolutional Networks (GCN) as linear propagation models and Graph Attention Networks (GAT) as non-linear propagation models. We focus on these two architectures because other models tend to exhibit similar behavior patterns. Unless stated otherwise, our default configuration employs a three-layer GNN with a feature dimension of $F = 25$ and utilizes the Leaky-ReLU activation function. For experiments involving full-rank matrices, we ensure that each layer's weight matrix is full rank, selected after an initial pre-training phase. Conversely, for studies with non-full-rank matrices, we set the rank loss parameter to a default value of 5. We adopt mean pooling as the default readout method throughout our experiments.

**Task Settings:** Our loss function is defined as $\|F_{\theta^*}(G_p) - C(G)\|$ for single-graph tasks, and $\sqrt{\sum_{G \in \mathcal{G}} \|F_{\theta^*}(G_p) - C(G)\|^2/M}$ for multi-graph tasks. Here $G_p$ is the combined graph with $G$ and graph prompt. Kindly note that $C(G)$ means an optimal function to the downstream task, which is not accessible without a specific task. Since the ultimate purpose of graph prompting is to approximate graph operation, we here treat $C(\cdot)$ as various graph data operations such as adding/deleting nodes, adding/deleting/changing edges, and transforming features of a given graph $G$. The intensity of these graph operations is controlled by a parameter $\beta \in [0, 1]$, where 0 indicates no change and 1 indicates generating a completely random new graph. In our experiments, we fix $\beta$ at 0.7, which means we have a 0.7 probability of removing a node/edge or masking some features.

*Definition of $C(G)$.* In our experiments, $C(G)$ represents the optimal graph-level embedding of a modified graph $G'$ derived from the original graph $G$. Specifically, $G'$ is obtained by performing a graph data manipulation operation on $G$, such as removing a certain percentage $\epsilon\%$ of edges or nodes. The function $C(G)$ is defined as:$C(G) = \text{Pooling}(\text{GNN}(G'))$ where GNN is a graph neural network, and Pooling is a graph-level pooling operation that aggregates node embeddings into a single vector representing the entire graph $G'$. This embedding $C(G)$ serves as the ground truth in our experiments.

*Experimental Procedure.* We focus on a graph-level task that inherently requires graph data manipulation. The experimental procedure is as follows:

1. **Graph Manipulation**: Starting with an original graph $G$, we create a modified graph $G'$ by randomly removing $\epsilon\%$ of edges or nodes. This simulates a data operation that alters the graph structure.

2. **Computing Ground Truth Embedding** $C(G)$: We pass $G'$ through a pre-trained and fixed GNN model followed by a pooling operation to obtain the graph-level embedding $C(G)$: $G' \rightarrow \text{GNN} \rightarrow \text{Pooling} \rightarrow C(G)$ This embedding represents the desired outcome of the data operation.

3. **Graph Prompting on** $G$: Instead of directly manipulating $G$, we apply a graph prompt $P_w$ to the original graph $G$ to approximate the effect of the manipulation: $G \rightarrow P_w(G)$ Here, $P_w(G)$ is the prompted graph, where $P_w$ is a learnable function parameterized by $w$.

4. **Computing Prompted Graph Embedding**: We pass the prompted graph $P_w(G)$ through the same GNN and pooling operations to obtain the embedding $F_\theta(P_w(G))$: $P_w(G) \rightarrow \text{GNN} \rightarrow \text{Pooling} \rightarrow F_\theta(P_w(G))$

5. **Error Computation**: We compute the error between the embeddings of the prompted graph and the ground truth embedding: Error $= \|F_\theta(P_w(G)) - C(G)\|$ This error quantifies how well the graph prompt approximates the desired graph data manipulation.

*Empirical Maximum Error in Figure 1.* Figure 1 in our paper illustrates the empirical maximum error observed in our experiments, corresponding to the theoretical upper bound discussed in Theorem 5. To approximate this upper bound: We perform multiple trials by repeating the experiment with different random removals of edges or nodes (i.e., generating different $G'$) and different initializations of the graph prompt $P_w$. For each trial, we compute the error as described above. We record the maximum error observed across all trials, which serves as an empirical approximation of the theoretical upper bound.

**Training:** In the graph prompting training process, we perform gradient descent on the parameters $\omega$ of the graph prompt $P_\omega$ using the Adam optimizer. We use a learning rate of $1 \times 10^{-4}$ and weight decay of $5 \times 10^{-5}$. We implement an early stopping mechanism with a maximum of 2000 epochs by default. When analyzing the upper bound of the error of the prompt method, it is crucial to ensure that the convergence value of $P$ is indeed the global minimum. To prevent the training of prompt parameters from falling into local minima, we approximate the global minimum by independently training $k$ times with random initialization for each prompt, and selecting the minimum loss. Typically, $k$ is set to 3.

**Testing:** Note that the error bound in Theorem 5 is the product of two terms: $\sin(\Phi/2)$ and $\|C(G)\|$. For a fixed pre-trained model, $\sin(\Phi/2)$ is consistent, but $C(G)$ varies with different generated datasets. In our experiments, the 30 graphs used to find the empirical maximum value are from the same dataset, while the different empirical maximum values are obtained from different datasets. Considering the average value better represents the change in the Error Bound relative to the independent variables like "Rank Lost".

**Codes:** https://anonymous.4open.science/r/dgpwadopwta/

