# OpenReview forum: "Does Graph Prompt Work? A Data Operation Perspective with Theoretical Analysis"
_ICLR.cc/2025/Conference — ICLR 2025 Conference Withdrawn Submission_

### Official Review · Reviewer_q53a · 2024-10-29

**Soundness:** 2
**Presentation:** 2
**Contribution:** 2
**Rating:** 3
**Confidence:** 3

**Summary:**

Graph prompting is a popular method to adapt pre-trained graph models for downstream tasks. While many graph prompting methods work well on real-world graph datasets, theoretical analyses on graph prompting methods are still not well explored. This study provides comprehensive analysis of popular graph prompting methods. The authors conduct extensive experiments on both synthetic and real-world datasets to validate the theoretical findings in this study.

**Strengths:**

1. The studied problem of theoretical analyses on graph prompts is a significant topic.
2. The authors provide experiments on both synthetic and real-world datasets.

**Weaknesses:**

1. Some arguments lack supporting evidence. The authors claim that “The rest graph prompt designs can usually be treated as their special cases or natural extensions.” in line 105. However, drawing this conclusion is not straightforward. For example, GPPT [1] designs graph prompts as structure tokens and task tokens, and GraphPrompt [2] designs graph prompts as prompt-based Readout operations. The authors may discuss how to treat these graph prompt methods as special cases or natural extensions of GPF or All-in-one.
2. Incorrect formulation of All-in-One. According to Section 2, All-in-One obtains the prompt graph by connecting $k$ learnable prompt tokens with $N$ original graph nodes. However, All-in-one in Lemma 1 adds prompt tokens to node features, which conflicts with Section 2. Since most theoretical analysis in this study is related to All-in-one, the authors should at least provide correct formulation of All-in-One as the basis of this paper.
3. The description of All-in-One-Plus in Section 4.1 is missing. The authors should provide the citation of this method and introduce it in the paper.
4. Notations are inconsistent in the paper, which confuses readers a lot. For example, the authors use two different notations $l$ and $i$ to represent the layer index of GNNs in the paper, while $l$ is also used to denote a column vector. The authors should use consistent notations to avoid ambiguity.
5. Duplicated sentences should be avoided. For example, the paragraph *Model Settings* appears twice in Section 5.1 and Appendix C.

[1] Sun, Mingchen, et al. "Gppt: Graph pre-training and prompt tuning to generalize graph neural networks." Proceedings of the 28th ACM SIGKDD Conference on Knowledge Discovery and Data Mining. 2022.
[2] Liu, Zemin, et al. "Graphprompt: Unifying pre-training and downstream tasks for graph neural networks." Proceedings of the ACM Web Conference. 2023.

**Questions:**

1. The authors take GCNs as linear graph models and GATs as nonlinear graph models based on different aggregation mechanisms. Could the authors provide some previous studies that use such categories? According to my knowledge, nonlinear graph models are typically GNNs with nonlinear activation functions between GNN layers.
2. What is meaning of $SG_\omega$ in Theorem 4?
3. What does $\mu$ mean in Theorem 5?
4. Could the authors provide more details about All-in-One-Plus?
5. Why do the authors not include the results of GPF-plus and All-in-one-plus in Section 5?

---

> ### Author Response · Authors · 2024-11-22
> **Rebuttal to q53a**
>
> We appreciate your thoughtful comments, and we feel sorry to see your negative score (ㅠ‸ㅠ) (ㅠ‸ㅠ) (ㅠ‸ㅠ)
>
> However, we firmly believe that you are such a nice reviewer because **most of your concerns are caused by some minor misunderstandings (especially W1 and W2)**. We felt your strong kindness from your comment like `The studied problem... is a significant topic`, and from your problems that are not tough for us to respond. To this end, we carefully prepared the following rebuttal to address your concerns comprehensively. **Hopefully they will change your mind and encourage you to raise your score, pushing this research forward.**

---

> > ### Author Response · Authors · 2024-11-22
> >
> > > W3: The description of All-in-One-Plus in Section 4.1 is missing. The authors should provide the citation of this method and introduce it in the paper.
> > >
> >
> > **R-3. All-in-One-Plus v.s. All-in-One**: In the paper of All-in-One, the authors designed an inserting pattern between the prompt and the original graph as some cross-link between prompt tokens and the original nodes. These cross-links are given by the dot product between prompt tokens and input graph nodes and then use a tailored connection. All-in-One-Plus removed tailored operation, which means all prompt tokens are connected to all original nodes with learnable connection weights. This is just a very natural extension of All-in-One and also included in their paper. All-in-One-Plus and All-in-One come from the same paper. To further address the reviewer’s concern, we added this introduction to our paper.
> >
> > ---
> >
> > > W4: Notations are inconsistent in the paper, which confuses readers a lot. For example, the authors use two different notations to represent the layer index of GNNs in the paper, while is also used to denote a column vector. The authors should use consistent notations to avoid ambiguity.
> > >
> >
> > **R-4**: We really thank the reviewer for this valuable suggestion!
> >
> > - We provide a glossary and default symbols meanings here for the reader’s convenience. Please see in Appendix A.1.2 (Table 2 and Table 3).
> > - To avoid ambiguity, each important formula is followed with a careful introduction of these notations. We try our best to ensure that our readers will not have notation misunderstanding **once they refer to the closest context.**
> > - **To further address the reviewer’s concern**, we followed the suggestion given by the reviewer and carefully checked our manuscript and updated notations.
> >
> > ---
> >
> > > W5: Duplicated sentences should be avoided. For example, the paragraph Model Settings appears twice in Section 5.1 and Appendix C.
> > >
> >
> > **R-5**: We really thank the reviewer for this valuable suggestion! Following the suggestion given by the reviewer, we re-write this paragraph in our Appendix and avoid duplicated sentences.

---

> > ### Author Response · Authors · 2024-11-22
> >
> > > Q1: The authors take GCNs as linear graph models and GATs as nonlinear graph models based on different aggregation mechanisms. Could the authors provide some previous studies that use such categories? According to my knowledge, nonlinear graph models are typically GNNs with nonlinear activation functions between GNN layers.
> > >
> >
> > **R-Q1:** We thank the reviewer for this interesting problem.
> >
> > - Kindly note that “linear/non-linear model” is not a special term for GNN. It is a very natural, very normal, and very comment word mathematically. For example, nonlinear activation functions between GNN layers are a non-linear component in GNN because they translate node features (say *X*) in a non-linear way. Similarly, for the message-passing stage (aggregation mechanisms) there also exists linear/non-linear ways for propagating node messages. For example, GCN has a linear aggregation w.r.t *X* (e.g. *X* = *AXW*) and GAT has non-linear aggregation. Technically, non-linear graph models contain two components that might indicate this nonlinearity: aggregation and activation.
> > - The big background of this paper is: that nearly all graph models contain nonlinear activation functions, but not every model has a nonlinear aggregation component. Our paper focuses on discussing the impact of linear/non-linear aggregations (GCN and GAT) beyond activation functions, therefore we think using “linear/nonlinear graph models” under the context/environment/setting of this paper should be elegant, concise, and clear without ambiguity. (It is a very important principle for a theory-intensive paper).
> > - To further address the reviewer’s concern, we replace the term “linear/nonlinear graph models” with “linear/nonlinear aggregation graph models” in any place that we think might cause misunderstanding.
> >
> > ---
> >
> > > Q2: What is meaning of SG_w in Theorem 4
> > >
> >
> > **R-Q2:** *SGw* is the prompt subgraph (including prompt token and token structure). The graph prompt changes the original graph *G* to *Gw* by integrating *G* with the prompt subgraph *SGw*. We thank the reviewer for pointing out this confused place and we have updated and added the explanation in our revised paper.
> >
> > ---
> >
> > > Q3: What does miu mean in Theorem 5
> > >
> >
> > **R-Q3:** Thank you for your insightful question regarding the meaning of *μ* in Theorem 5.
> >
> > - In the proof of Theorem 5, we perform an in-equation scaling analysis that allows us to bound the error as the product of two components. The first component depends solely on the model parameters and reflects the model’s characteristics. The second component is related to the norm of the graph embedding vector *C*(*G*), which is intrinsic to the graph itself.
> > - The parameter *μ* is a function of the model parameters *θ* and represents the specific upper bound of the component that depends only on the model. Since this upper bound is uniquely determined by the various parameters of the model, μ can be regarded as an implicit function of *θ*.
> > - In the subsequent paragraphs of our paper like equation (4) and Appendix A.3.3), we provide the explicit form of this implicit function *μ*. Intuitively, deriving *μ* involves analyzing the expressive power of the model and understanding the corresponding structure of the embedding space.
> >
> > We hope this explanation clarifies the role of *μ* in Theorem 5. Please let us know if you have any further questions or need additional clarification.
> >
> > ---
> >
> > > Q4: Could the authors provide more details about All-in-One-Plus?
> > >
> >
> > Please see our response to W3.

---

> > ### Author Response · Authors · 2024-11-22
> >
> > > Q5: Why do the authors not include the results of GPF-plus and All-in-one-plus in Section 5?
> > >
> >
> > **R-Q5:** Thank the reviewer for this question:
> >
> > - **Why All-in-One/GPF-Plus in Figure 1?** We do this **JUST** to confirm the following claim in our paper (see in Line 291-292) “*…more advanced graph prompts… generally have a lower bound than the naive one…*”. This is a natural observation drawn from our theory analysis in section 4.1. We could have come to this observation by just simply comparing All-in-One and GPF, however, since they belong to two prompt categories (prompt as tokens, and prompt as a graph, with different inserting patterns), we think it would be more convincing to compare within different groups (GPF v.s. GPF-Plus, and All-in-One v.s. All-in-One-Plus).
> > - **Why no All-in-One/GPF-Plus in others?** From a theoretical perspective, there is no fundamental difference between these variants to their regular models. **Kindly note that our experiments are strictly corresponding to each Theorem we deduced,** all of which are built upon All-in-One and GPF. In particular, section 5.2 corresponds to Theorems 3, 4, and 9. Section 5.3 corresponds to Theorem 5. Section 5.4 corresponds to Theorem 6 and Theorem 7. Figure 2 corresponds to Theorem 8. We do this because we wish to **clearly present** the consistency between our theory and experiment to our readers, without too many trivial distractions. It is a very important principle for a theory-intensive paper to reflect their theory in nature.
> > - **We Act:** To further address the reviewer’s concern, we supplemented more experiments regarding All-in-One-Plus/GPF-Plus in our open code project. Please check this URL https://anonymous.4open.science/r/dgpwadopwta/supplement.pdf, from which we can see that there are no inconsistent observations that conflict with our theory analysis.

---

> ### Author Response · Authors · 2024-11-22
>
> > W1: Some arguments lack supporting evidence. The authors claim that “The rest graph prompt designs can usually be treated as their special cases or natural extensions.” in line 105. However, drawing this conclusion is not straightforward. For example, GPPT [1] designs graph prompts as structure tokens and task tokens, and GraphPrompt [2] designs graph prompts as prompt-based Readout operations. The authors may discuss how to treat these graph prompt methods as special cases or natural extensions of GPF or All-in-one.
> >
>
> **R-1.1. GPPT:** In GPPT, the structure tokens can be treated as some prompt tokens in All-in-One, and the task tokens in GPPT represent node category, which are also some special tokens of All-in-One’s prompt graph. Their task can be treated as predicting part of token structure between structure tokens and task tokens to achieve node classification.
>
> The above observation is also confirmed by the authors of All-in-One in their paper [1]. We kindly encourage the reviewer to check in section 3.5.1 of their paper. Here we briefly copied their content for your information:
>
> > Their work somehow is a special case of our method when our prompt graph only contains isolated tokens, each of which corresponds to a node category.
> >
>
> **R-1.2. GraphPrompt:** In most cases, prompt tokens are expected to be inserted in the node features, which can be treated as some operation at the “0-th” layer of the graph models. Our theory discussed how “[1,end]” layers of the GNN act with graph prompts. Since graph pre-trained models are frozen in the graph prompting area, we can also extend the discussion by changing the position of the graph prompt to *k*-th layer of the GNN where [1, *k*) layers are treated as fixed feature mapping function and (*k* − *end*] layers still hold the findings of our theory framework. From this perspective, GraphPrompt, obviously, can be treated as some special case of this paper and our analysis framework is applicable to this model.
>
> **R1.3. Other Designs:** For more kinds of graph prompt models, Sun et al. [2] have carefully discussed their inner connections to All-in-One. Please see Section 5.4 of this survey [2] for your information.
>
> We would very much like to add these discussions to our paper. However, these opinions have been carefully discussed in a very comprehensive survey and many detailed papers. So we think it might be a little bloated and redundant for our goal that focus on concisely discussing an elegant theory. **To further address the reviewer’s concerns**, we cite these mentioned papers in the mentioned sentence “The rest graph prompt designs can usually be treated as their special cases or natural extensions” in our paper.
>
> Ref:
>
> 1. Xiangguo Sun, Hong Cheng, Jia Li, Bo Liu, Jihong Guan. All in One: Multi-task Prompting for Graph Neural Networks. KDD 2023
> 2. Xiangguo Sun, Jiawen Zhang, Xixi Wu, Hong Cheng, Yun Xiong, Jia Li. Graph Prompt Learning: A Comprehensive Survey and Beyond. https://arxiv.org/abs/2311.16534

---

> ### Author Response · Authors · 2024-11-22
>
> > W2: Incorrect formulation of All-in-One. According to Section 2, All-in-One obtains the prompt graph by connecting learnable prompt tokens with original graph nodes. However, All-in-one in Lemma 1 adds prompt tokens to node features, which conflicts with Section 2. Since most theoretical analysis in this study is related to All-in-one, the authors should at least provide the correct formulation of All-in-One as the basis of this paper.
> >
>
> **R-2 (They Are Consistent):** We thank the reviewer for this comment.
>
> - **Section 2** described All-in-One as a general format.
> - **Lemma 1** presented the transformation of the graph after the prompt. Please kindly note what we said (lines 853-854 in Lemma 1): “*Without loss of generality, we assume that the prompt subgraph in All-in-One has only one node.*”
> - **Why?** Because we said (line 862): “*For All-in-One, the prompt subgraph is connected to graph a parameterized way, i.e. there exist parameters that control the connection relationship between **any two nodes** in the prompt subgraph and the original graph.*”
> - **Conclusion:** Lemma 1 presents how feature transformed from any (token, node) pair. Equations (23-24) present what happened in inner graph models by adding a graph prompt (All-in-One). We do this to reduce the understanding bar for our readers.
>
> To further address the reviewer’s concern, we have double-checked and updated anything we can do for more readable content.

---

> ### Author Response · Authors · 2024-11-22
>
> We thank the reviewer for your comment. According to other reviewers’ scores (8 and 6), ``your opinion is very important to us!`` Please let us know if you have any further questions or suggestions. We are warmly looking forward to your response and we are glad to discuss further with you!
>
> Kind regards.

---

> ### Comment · Reviewer_q53a · 2024-11-23
> **Reviewer Feedback to Author Response**
>
> I appreciate the detailed feedback from the authors. However, I am afraid that my concerns are still not addressed. I got some new questions from the authors' feedback, but I first want to confirm the correctness of formulating All-in-One in Section 2 and Lemma 1.
>
> In Section 2, we know that the prompted graph $\mathcal{G}\_\omega$ will have $N+k$ nodes, i.e., $N$ original nodes and $k$ token nodes. When we assume only one prompt token, $\mathcal{G}\_\omega$ will have $N+1$ nodes.
> - NQ1: could the authors specify the shape of matrix $\mathbf{A}\_\omega$ in line 118?
>
> In Lemma 1, we compute $S\_\omega$ and $X\_\omega$ for All-in-One using Equation (23). Here, $S\_\omega$ is the diffusion matrix of $\mathcal{G}\_\omega$ and $X\_\omega$ is the node feature matrix of $\mathcal{G}\_\omega$.
> - NQ2: what does $l \in \mathbb{R}^{N-1}$ represent in Lemma 1?
> - NQ3: could the authors specify the shape of $S\_\omega$ and $X\_\omega$ in Equation (23)?
>
> Thanks,
> Reviewer q53a

---

> > ### Author Response · Authors · 2024-11-25
> >
> > Dear Reviewer,
> >
> > Thank you for your careful examination of our matrix formulations. Let me clarify the dimensional relationships and notation used in Section 2 and Lemma 1.
> >
> >
> > 1. **About the $A_w$ Matrix (NQ1)**
> >    - Indeed, when adding one prompt token to a graph with N original nodes, the prompted graph's adjacency matrix $A_w$ should be (N+1)×(N+1)
> >
> > 2. **About Lemma 1 Notation (NQ2 & NQ3)**: To maintain consistency across different prompting methods, we use a slight notation shift:
> >    - We denote the original graph size as (N-1) nodes when discussing All-in-One
> >    - After adding the prompt node, we get N×N matrices
> >    - Vector $l$ represents the aggregation weights from the prompt node to the original (N-1) nodes,
> >    - Therefore, $S_w$ and $X_w$ in Equation (23) are N×N matrices
> >
> > In our revised version (line 868), we carefully mark this slight shift to our readers and please kindly note that this shift is a normal math trick, which will not impact the math conclusion but can reduce math complexity.
> >
> > 3. **Notation Consistency**: While it might seem more intuitive to use (N+1)×(N+1) for All-in-One and N×N for GPF. We chose to use N×N consistently for All-in-One by shifting the base notation (using N-1 as the original node count). This shift is merely notational and doesn't affect the mathematical validity
> >
> >
> > Thanks again for your interest in our work! and please feel free to let us know if you have any further suggestions/questions.

---

> > > ### Comment · Reviewer_q53a · 2024-11-25
> > >
> > > Thanks for your reply. When we denote $N-1$ nodes for All-in-One, the node feature matrix $X$ should have $N-1$ rows, right? So the question is: how can we obtain an $N$-row matrix $X_\omega$ by $X_\omega=X+\textbf{e}_N \textbf{p}^\top$ in Equation (23)?

---

> > > > ### Author Response · Authors · 2024-11-26
> > > >
> > > > Dear Reviewer q53a
> > > >
> > > > We truly appreciate your time in reviewing this work! We write to ask whether you have any further questions. Please do not hesitate to let us know since phase 1 will soon come to a close.
> > > >
> > > > We humbly wish you could reconsider our work if the mentioned misunderstandings are clarified. **Your opinion is very important to us and we humbly hope that you would view our work favorably** and consider raising your scores to reflect the improvements made. We carefully checked all your questions and we have found that nearly all your questions/weaknesses are largely neutral without too obvious negative intent. We believe our work deserves a more favorable score if your misunderstandings are clarified.
> > > >
> > > > Graph prompts have recently been widely treated as a very promising way at the data level towards more general graph-based AI applications. However, despite its potential, ``the theoretical foundation of graph prompting is nearly empty, raising critical questions about its fundamental effectiveness.`` The lack of rigorous theoretical proof of why and how much it works is more like a **“dark cloud”** over the graph prompting area to go further. **And that is why we are truly proud of this work as it advances the theoretical understanding of graph prompting and redefines previous efforts in the field.**
> > > >
> > > >
> > > > Currently, the research community is filled with too many empirical studies but we urgently need to figure out these foundational theories to support us to go further. Our contributions provide a solid foundation for future explorations and can significantly benefit practical applications by offering insights into the capabilities and limitations of graph prompting techniques.
> > > >
> > > >
> > > > Thanks again for your time. We are warmly looking forward to your letter.
> > > >
> > > > Kind regards

---

> > > > > ### Comment · Reviewer_q53a · 2024-11-26
> > > > >
> > > > > Thanks for your reply.
> > > > >
> > > > > When we set $X$ by using the first $N-1$ rows for original $|\mathcal{V}|=N-1$ graph nodes with a zero vector as the last row, computing $X_\omega$ in Equation (23) means we replace the last row of $X$ with the prompt vector $p$, given that $e_N$ is a one-hot vector $[0, 0, \cdots, 1]^\top$ according to line 868. Therefore, All-in-One in Lemma 1 will convert the original graph $\mathcal{G}$ with $N-1$ nodes to a prompted graph $\mathcal{G}_\omega$ with $N$ nodes by adding a new prompt node to the original graph, while the feature vector of each graph node is unchanged. However, I am afraid that this formulation is inconsistent with the design in All-in-One [1]. According to Section 3.3.4 in All-in-One [1], the inserting pattern is defined as adding prompt tokens to node features: $\hat{x}_i = x_i+\sum\_{k=1}^{|\mathcal{P}|} w\_{ik} p_k$ ($|\mathcal{P}|=1$ in your setting). Therefore, All-in-One should alter the feature matrix of the graph nodes. Could you help to clarify this?
> > > > >
> > > > > [1] Sun, Xiangguo, et al. "All in one: Multi-task prompting for graph neural networks." KDD. 2023.

---

> > > > > > ### Author Response · Authors · 2024-11-27
> > > > > >
> > > > > > Very good question! We thank you for your further reply, which finally clarifies what exactly confused you! Now, let's clarify this question, and hopefully, this time, we can finally clear up your confusion!
> > > > > >
> > > > > >
> > > > > > - You can treat $X_w$ as a collection of features from original nodes and the prompt tokens. However,$X_w$ itself can not define a detailed inserting pattern. The inserting pattern needs to be defined within $S_w$ (see equation 23), from which we can see that $l$ defines how to connect the prompt token to other original nodes (inserting pattern). and $S_{NN}$ defines how to organize the inner structure within a graph prompt (here we have one token).
> > > > > >
> > > > > > - We think the reviewer might have some misunderstanding of the paper All-in-One because the original content of this paper said:
> > > > > > > We can define the inserting pattern as the dot product between prompt tokens and input graph nodes, and then use a tailored connection...
> > > > > >
> > > > > > - Here $l$ can be treated as such dot product, of course, it can also be treated as free parameters. (Dot produce is just one example given by the author of that paper, not the only way)
> > > > > >
> > > > > > - We also dive into the open code of the paper All-in-One and ensure that its public implementation does not conflict with our formula.
> > > > > >
> > > > > >
> > > > > > - To further address the reviewer's concern, we sent an email to the author of All-in-One and asked them to help us double-check this issue, and we got clear confirmation from the authors of that paper.

---

> > > > > > > ### Comment · Reviewer_q53a · 2024-11-29
> > > > > > >
> > > > > > > Thanks for your clarification.
> > > > > > >
> > > > > > > In the paper of All-in-One, the authors provide only one way to obtain $X_w$ by $\hat{x}_i = x_i+\sum\_{k=1}^{|\mathcal{P}|} w\_{ik} p_k$ with its simplified version. So I think the theoretical analysis should follow the default inserting pattern. The current formulation of the inserting pattern in your submission does not appear in the All-in-One paper, which will make readers hard to follow your analysis.
> > > > > > >
> > > > > > > Additionally, you mentioned that the open code of the All-in-One paper does not conflict with your formulation. I have not checked their implementation yet. Do you mean the inserting pattern in their code is inconsistent with the presented one in the All-in-One paper?

---

> > > > > > > > ### Author Response · Authors · 2024-11-29
> > > > > > > >
> > > > > > > > Dear Reviewer q53a
> > > > > > > >
> > > > > > > > I think you might have some misunderstanding on their paper (or largely caused by their presentation, not ours). If you look at their paper, they said:
> > > > > > > >
> > > > > > > > > We can define the inserting pattern as the **dot product between prompt tokens and input graph nodes, and then use a tailored connection** like
> > > > > > > > $ \mathbf{\hat{x}}_i=\mathbf{x}_i+\sum_{k=1}^{|\mathcal{P}|} w_{ik}\mathbf{p}_{k} $ where $ w_{ik} $ is a weighted value to prune unnecessary connections
> > > > > > > >
> > > > > > > >
> > > > > > > >
> > > > > > > > The above content truly has some confusing things for readers, so we understand your misunderstanding and we initially have a similar confusion with you. However, after carefully checking their paper, their code, and a **face-to-face online meeting** with the original authors of that paper, we can ensure that:
> > > > > > > >
> > > > > > > > This equation does not denote how to insert a prompt token to the graph, it presents when (or after) we connect a prompt token to a node (via cross-links), we send this combined graph to a GNN, and this equation is an example how GNN aggregates a node's feature via its neighbors with prompted graph, and the above equation talks how this feature changed when we sent this combined graph into a GNN. All-in-One does not directly change node features but inserts the prompt to the original graph, and the feature changing happens within a GNN.
> > > > > > > >
> > > > > > > > ``One More Thing``:
> > > > > > > > We have realized that **nearly all your misunderstandings and nearly all your proposed weaknesses of our paper, are based on your misunderstanding** of ``THAT paper``, not ours! Instead, our presentation in our paper more clearly presents the nature of All-in-One (We did a better job than that paper's expression). We feel sorry that THAT paper did not clearly present its idea. But, we wish kindly to say: **THIS is not our fault.**
> > > > > > > >
> > > > > > > > We have carefully checked this issue again and again, with the original authors, with their papers, with their codes, and with everyone we can turn to help.
> > > > > > > >
> > > > > > > >
> > > > > > > > This year, ICLR is more competitive than ever. We do hope our huge effort could address your question and we are looking forward to your further raising. Feel free to let us know if you have any further questions.
> > > > > > > >
> > > > > > > >
> > > > > > > > Kind regards,

---

> > > > > > > > > ### Comment · Reviewer_q53a · 2024-11-30
> > > > > > > > >
> > > > > > > > > Thanks for your reply.
> > > > > > > > >
> > > > > > > > > I would like to briefly clarify why I care about the formulation of All-in-One in your paper so much. You paper aims to do theoretical analysis of graph prompting studies, especifically for All-in-One and GPF. We can say at least 50% of your paper is related to or based on All-in-One. So the basic requirement of this study is to correctly formulate All-in-One and make it clear to readers. For example, when the construction of $X$ in Equation (23) and $e_N$ is not specified in your initial submission, readers may have trouble understanding Equation (23) and following the subsequent analysis.
> > > > > > > > >
> > > > > > > > > While I still think the inserting pattern introduced in the All-in-One paper is modifying node features with prompt nodes, I am now clear with your formulation in Equation (23) based on the revised PDF. Since the discussion period is ending soon, I think we may set it aside and move forward.
> > > > > > > > >
> > > > > > > > > In the revised PDF, the authors mentioned that "All-in-One-Plus treats the inserting pattern as learnable weights". Could you provide more detailed explanation about this? The authors provide formal formulation of GPF, GPF-plus, and All-in-One in the paper but not for All-in-One-Plus yet.

---

> > > > > > > > > > ### Author Response · Authors · 2024-11-30
> > > > > > > > > >
> > > > > > > > > > Dear Reviewer q53a:
> > > > > > > > > >
> > > > > > > > > > Thanks for your further discussion.
> > > > > > > > > >
> > > > > > > > > > > I would like to briefly clarify why I care about the formulation of All-in-One in your paper so much...While I still think the inserting pattern introduced in the All-in-One paper is modifying node features with prompt nodes, I am now clear with your formulation..., I think we may set it aside and move forward...
> > > > > > > > > >
> > > > > > > > > >
> > > > > > > > > > **R1**: We totally understand and that is the reason why we explain this to you again and again with patience. However, please kindly note that **your understanding of All-in-One is Wrong**. ``It is unfair to give us a negative impression based on the wrong understanding of other papers``, which is largely caused by their presentation, not ours (Although we totally understand your misunderstandings and fully respect you).
> > > > > > > > > >
> > > > > > > > > > We did a better job than the All-in-One paper with a more concise and clear presentation of the nature of All-in-One. And that is also an unexpected contribution: ``For many other readers to All-in-One like you, our paper may help them clear up their confusion``. This is very meaningful because **All-in-One is one of the most classic, most fundamental, and most straightforward models in this emerging area.** Anyone who knows something about graph prompts should apparently know it well, which can be even treated as a bar for this area researchers.
> > > > > > > > > >
> > > > > > > > > > We are proud that our work may help the community understand it better, clearing up some potential misunderstandings about All-in-One. We thank you for setting this issue aside and moving forward. Trust me, you won't regret it.
> > > > > > > > > >
> > > > > > > > > > ---
> > > > > > > > > >
> > > > > > > > > >
> > > > > > > > > > > In the revised PDF, the authors mentioned that "All-in-One-Plus treats the inserting pattern as learnable weights". Could you provide a more detailed explanation about this? The authors provide a formal formulation of GPF, GPF-plus, and All-in-One in the paper but not for All-in-One-Plus yet.
> > > > > > > > > >
> > > > > > > > > >
> > > > > > > > > > **R2**: You asked about similar problems before, please check our responses to your W3 and Q5. In addition, we copied our response to your previous questions here to help you understand more details on this: "All-in-One-Plus treats the inserting pattern as learnable weights":
> > > > > > > > > >
> > > > > > > > > > The inserting pattern needs to be defined within $S_w$ (see equation 23), from which we can see that $l$ defines how to connect the prompt token to other original nodes (inserting pattern). and $S_{NN}$ defines how to organize the inner structure within a graph prompt (here we have one token). Here $l$ can be treated as such dot product (defined in All-in-One as their inserting pattern), of course, it can also be treated as free parameters (All-in-One Plus as mentioned in our paper).
> > > > > > > > > >
> > > > > > > > > >
> > > > > > > > > >
> > > > > > > > > > Feel free to let us know if you have any further question.
> > > > > > > > > >
> > > > > > > > > > KInd regards.

---

> > > > > > > > > > ### Author Response · Authors · 2024-12-01
> > > > > > > > > >
> > > > > > > > > > Dear Reviewer q53a
> > > > > > > > > >
> > > > > > > > > >
> > > > > > > > > > The discussion period is ending soon. Please let us know if we have solved your question.
> > > > > > > > > >
> > > > > > > > > >
> > > > > > > > > > Kind regards,

---

> > > > > > ### Author Response · Authors · 2024-11-28
> > > > > >
> > > > > > Reviewer q53a
> > > > > >
> > > > > > We truly thank you for your kind engagement in our discussion. We kindly inquire if your previous questions have been clarified and please do not hesitate to contact us if you have any further questions or suggestions.
> > > > > >
> > > > > > We value your opinion and please do not hesitate to let us know if our previous effort changed your mind.
> > > > > >
> > > > > > We are warmly looking forward to hearing the good news from you, and we are also glad to discuss further with you! Please do not hesitate to give us any further feedback at your earliest convenience.
> > > > > >
> > > > > >
> > > > > >
> > > > > > Kind regards.

---

> > > > > > ### Author Response · Authors · 2024-11-29
> > > > > >
> > > > > > Dear **Reviewer q53a**
> > > > > >
> > > > > > Today is Thanksgiving, a day to express our heartfelt thanks. We are grateful to every reviewer for giving their time in this phase and for walking alongside our paper. In the past few days, may everything we encounter become part of life’s beautiful scenery.
> > > > > >
> > > > > > We hope our efforts in the past few days have clarified all your questions. Please do not hesitate to let us know if our previous efforts changed your mind positively.
> > > > > >
> > > > > > We are glad to discuss further with you! Please do not hesitate to give us further feedback at your earliest convenience.
> > > > > >
> > > > > > Kind regards.

---

> ### Author Response · Authors · 2024-11-25
>
> Kindly note that $X \in \mathbb{R}^{N \times F }$ here is the natural extension of node attribute matrix $X_{0}\in \mathbb{R}^{N-1 \times F }$ by adding an additional zero vector $ \mathbf{0}_F \in \mathbb{R}^{1 \times F } $ like:
>
> \begin{bmatrix} X_{0} \\\\ \mathbf{0}_F \end{bmatrix}
>
>
>
> the display of the above equation might not work in this openreview system because it can not support a math array very well.  **We have supplemented the above-detailed descriptions in the appendix. Please check in lines 868-872.**
>
> This treatment ensures dimensional consistency while maintaining the conciseness and uniformity of the expression without affecting the final computational results.

---

### Official Review · Reviewer_sjg4 · 2024-10-31

**Soundness:** 3
**Presentation:** 2
**Contribution:** 3
**Rating:** 5
**Confidence:** 3

**Summary:**

This study provides a solid theoretical analysis of graph prompts. The theoretical findings include the capabilities of graph prompts on GCN models with non-linear layers, the error bound of the data operations by graph prompts for both a single graph and batch of graphs, and the error distributions of the data operations by graph prompts. This work also provides empirical studies to confirm these theoretical findings.

**Strengths:**

The theoretical analysis in this study addresses the gap in establishing a theoretical basis for the capabilities of graph prompts with non-linear pretrained models and training on batches of graphs. The theoretical findings demonstrate the capabilities of graph prompts across several typical GNN models, providing detailed error bounds and error distributions. Notably, the authors show that the scale of prompts does not increase linearly with the size of the graph dataset, a positive result that supports the scalability of graph prompts.

**Weaknesses:**

The writing could be further improved to assist readers who may not have sufficient background knowledge about graph prompts. It also lacks some explanations, which might lead to misunderstandings among readers. For instance:

1. The description of the function $C$ is vague and could be clarified using a specific task, such as binary classification. Additionally, $C$ should be denoted as a function of a certain downstream task.
2. The terms  $\Phi ,  \mu , and  \lambda$  in Equation (4) are not explained and should be clarified.
3. The precondition in Corollary 1 is not specified and should be stated explicitly.

**Questions:**

1. Why is the error related to the prompt design in Theorem 5? Intuitively, the term $||C(G)||$ appears to be related to the graph itself and the downstream task, suggesting it may not depend with the prompt design.
2. Does the number of non-linear layers affect the error bound in Theorem 5?
3. How is $C(G)$ computed in Figure 1?

---

> ### Author Response · Authors · 2024-11-22
> **Rebuttal to sjg4**
>
> **We are truly grateful for your support to our work, and we’re moved to tears!**  (╥﹏╥)  (╥﹏╥)  (╥﹏╥)
>
> We hope below responses could address your questions and **encourage you to further champion and fight for our paper in the later reviewer discussion phase.**

---

> ### Author Response · Authors · 2024-11-22
>
> > W1: The description of the function C is vague and could be clarified using a specific task, such as binary classification. Additionally, C should be denoted as a function of a certain downstream task.
> >
>
> **R-1:** We appreciate the reviewer’s suggestion about the clarity of *C*
>
> - In our paper, we defined $C$ as a mapping function that maps the original graph $G_{ori}$ to embedding $C(G_{ori})$, which achieves good performance in downstream tasks.
> - To make this concept more concrete, we can illustrate through a binary classification task: Consider an optimal model $F$ that accurately performs binary classification on graphs, consisting of an encoder $E$ and a task head. The encoder $E$, which maps graphs to vector space, can be viewed as a specific implementation of the function $C$, here $C$ mostly refers to potential solutions to downstream tasks, which are usually unseen. We can treat it as some implicit function to the downstream task.
> - However, there are lots of downstream tasks, making further discussing detailed $C$ impractical and very hard. Our paper focuses on the nature of graph prompts in manipulating graph data. However, the capability of graph data manipulation is not always needed in graph-based applications. Figuring out this is far out of the scope of this paper. Therefore, we avoid being trapped in detailed tasks so that we can **clearly present** the nature of graph prompt in theory to our readers, without too many trivial distractions. We believe it is a very important principle for a theory-intensive paper to reflect their theory in nature.
> - To further reduce the bar of understanding for our reader, we follow the suggestion given by the reviewer and provide more explanation/discussion on $C$ in our paper.
>
> ---
>
> > W2: The terms Φ, μ, andλ in Equation (4) are not explained and should be clarified.
> >
>
> **R-2:** Let us clarify the roles of *Φ*, *μ*, and *λ* in Equation (4):
>
> - Regarding *μ* and *λ*: As mentioned in line 266, these terms represent a decomposition of the error into two multiplicative components. *μ* corresponds to model-dependent factors, depending solely on model parameters. *λ* corresponds to graph-dependent factors (given a fixed downstream task). We use “correspond to the model and graph” deliberately, as these are not explicit functions with definite computational procedures, but rather theoretical constructs representing these dependencies.
> - Regarding *Φ*: This is an angle measurement that solely depends on model parameters. It characterizes the landscape (expression space) formed by the embedding vectors when varying prompt parameters. Specifically, *Φ*/2 provides an upper bound for the angle between the target embedding *C*(*G*) and the expression space. Intuitively, a more expressive model leads to a more flexible expression space, which results in smaller angles and thus tighter upper bounds. To reduce the understanding bar for our readers, we use the phrase “a measurement of the model’s expressiveness” to capture this intuition in a more accessible way.
> - We give more details on these in Appendix A 3.3. We hope this explanation clarifies these. Please let us know if you have any further questions or need additional clarification.
>
> ---
>
> > W3: The precondition in Corollary 1 is not specified and should be stated explicitly.
> >
>
> **R-3:** We thank the reviewer for this problem. Kindly note the *ϵ* in Corollary 1 follows exactly the same definition and conditions as in Theorem 8. This corollary is derived directly from Theorem 8, and it is a basic quality of Chi distribution.

---

> ### Author Response · Authors · 2024-11-22
>
> > Q1: Why is the error related to the prompt design in Theorem 5? Intuitively, the term $||C(G)||$ appears to be related to the graph itself and the downstream task, suggesting it may not depend with the prompt design.
> >
>
> **R-Q1:** Thanks for your question. Kindly note that $||C(G)||$ is talking about the **upper bound** of the error, not the error itself. Since $C$ is usually unseen (an optimal solution to the downstream task), and the graph prompt aims to approximate $C(G)$ by $P_w(G)$ and the frozen graph model $F_{\theta}$, leading to $||F_{\theta}(P_w(G))|| \rightarrow ||C(G)||$. That means (although the upper bound of the error is related to the graph itself and the downstream task) the error itself in the graph prompting setting is also related to the prompt design in practice. We draw this discussion just to indicate that some intuitional findings from previous empirical work like the paper All-in-One are reasonable. In that paper, the authors empirically observed that the error of graph prompts in the approximation of graph data manipulations may relate to the non-linear layers of the graph model and the prompt design (please see our motivation lines 126-128 for your information), which now has more solid evidence from our theory.
>
> ---
>
> > Q2: Does the number of non-linear layers affect the error bound in Theorem 5?
> >
>
> **R-Q2:** Yes, it does. This is a very interesting and insightful question, which is informative to guide real engineering practice. We believe that if we didn’t offer our theory, answering this question would be very hard and intractable. But now, anyone who carefully reads our paper can systematically answer this question (although it might deserve a new research paper to fully answer this question by our theory, and we encourage other researchers/engineers to follow our work for further studying this question ). Here we give a basic analysis of this problem with the help of our theory (e.g. Theorem 5): The number of non-linear layers influences the error bound through its effect on *Φ* via two competing effects:
>
> - Positive Effect: Increasing non-linear layers may enhance model expressiveness. This can lead to a decrease in the angle *Φ*/2, potentially resulting in a tighter error bound.
> - Counter Effect: With increased model expressiveness, $F_{\theta^*}(G)$ evolve, the gap between the two embeddings, $F_{\theta^*}(G)$ and $C(G)$, might increase, which could lead to a larger angle $\Phi/2$.
> - Empirical Support: Our experimental results (Figure 4) in the figures support the interplay between these competing effects.
>
> ---
>
> > Q3: How is C(G) computed in Figure 1?
> >
>
> **R-Q3:** *C*(*G*) means an optimal function to the downstream task, which is not accessible without a specific task. Since the ultimate purpose of graph prompting is to approximate graph operation, we here treat *C*(⋅) as various graph data permutations such as adding/deleting nodes, adding/deleting/changing edges, and transforming features of a given graph *G*. Then we wish to see how well the graph prompt reaches *C*(*G*) by manipulating graph data with a graph prompt. Then *C*(*G*) can be treated as graph-level embedding after we change the given graph *G*.

---

> > ### Comment · Reviewer_sjg4 · 2024-11-23
> >
> > I appreciate your efforts in providing further clarification. However, some of my concerns remain unaddressed, and I will elaborate on them in detail below.
> >
> > **Regarding W1.** If I understand correctly, $C(G)$ could be expressed as two-dimensional embeddings for nodes in binary classification problems. Specifically, if node $i$ belongs to class 0, we have $C(G)[i]=(1,0)$; otherwise, $C(G)[i]=(0,1)$. It will improve readability if you clearly explain $C(G)$ in a similar way.
> >
> > **Regarding W2.** $\lambda$ and $\mu$ are commonly used to denote the eigenvalues of a graph Laplacian and an adjacency matrix in graph learning literature. However, $\lambda$ in this paper refers to an intermediate value used in your proof.  I recommend using other notations to represent the errors of two parts, and more details should be provided to clarify these errors in the main text.
> >
> > **Regarding Q3.** I still do not fully understand how the error in Figure 1 is computed. The error represents the gap between the embeddings of the modified graph after graph prompting and the ground-truth $C(G)$. Is $C(G)$ the answer to the downstream task as I speculated in *Regarding W1*? Additionally, what specific downstream task and dataset are used in Figure 1?
> >
> > Overall, the presentation of your work needs improvement to help readers fully understand the concepts and results. Kindly note that you can upload a revised version of your paper during the ICLR rebuttal period to provide further clarification for me and other reviewers.

---

> ### Author Response · Authors · 2024-11-25
>
> Dear Reviewer,
>
> Thanks for giving us time to clarify your question further!
>
>
>
> **Regarding W1 and Q3:** Yes, your understanding is almost correct. The only minor difference is that we treat $C(G)$ as embeddings instead of direct results. That means the downstream task should have its own task head to extract a correct answer from $C(G)$. **Take the graph classification as an example**:
>
> $G \rightarrow GNN \rightarrow pooling \rightarrow g^{'} \rightarrow task head \rightarrow results$
>  (apparently your original understanding is a special case of this one)
>
> where $g^{'}$ is a graph-level embedding and our paper claims that there exist an optimal embedding denoted by $C(G)$ making downstream task perform well. Apparently, without knowing exact task head, we never know such optimal $C(G)$. In practice, we can get sub-optimal $C(G)$ by interatively tuning task head and front models. And more frequently in graph prompting area, both GNN and task head are given and fixed, in this case, we can find $C(G)$ by minizing target loss at the optimal point.
>
>
>
> **How about node classification?** The whole discussion of this paper is built upon graph-level tasks since we said in lines 96-99 that:
>
> > For node-level and edge-level tasks, many studies (Sun et al., 2023a; Liu et al., 2023) have proved that we can always
> find solutions to translate these tasks to the graph-level task
>
> By concentrating on graph-level tasks, we provide a clearer evaluation of the effectiveness of graph prompting in approximating graph transformations.
>
> **Is graph manipulation always helpful for any downstream task / dataset?**
>
> In this paper, we argue the poweful capability of graph prompt in manipuating graph data. However, is graph manipulation always helpful for any kind of downstream task (or dataset)? That is an open problem needing to be answered and that might be the reason why graph prompts sometimes do not "work well" in some cases as reported by some empirical study. Figuring out this problem deserve a hundred of new research papers and application studies. To this end, we avoid discussing detailed downstream task. Instead, we focus on one "special" task, the performance of which nearly defininetly relate to data manipuation. This task is:
>
> > changing part of edges/nodes etc from graph $G$ and get a new graph $G^’$, then we have two graphs, $G$ and $G^{’}$. The task target is to find the graph embedding of this manipulated graph $G^{’}$.
>
> In this task, $C(G)$ means graph-level embedding of $G^{’}$ (ground truth and we can easily get this ground truth by simply calculating the pooling of the manipulated graph). The we consider how well does graph prompt approximate this ground truth and report the error between them.
>
> However, the above error is just one normal error value, not the upper bound, not the error distribution. Figure 1 should be responsible for Theorem 5, which studies the error upper bound. To this end, we randomly repeat the above setting several times and choose the max error as an approximation of the upper bound from Theorem 5, and that is the reason why we name the vertical axis of Figure 1 as "empirical max error".
>
>
> **Regarding W1, W2 and  Q3:**
>
> We truly thank the reviewer for pointing out these questions. To make the paper more readable, we polish the above explanation in our revised paper (see Appendix C on page 33).

---

> > ### Comment · Reviewer_sjg4 · 2024-11-26
> >
> > Thank you for your further clarification. Regarding Figure 1, if I understand correctly, you change the graph structure and features of $G$ randomly and then set the loss function as the distance between the embeddings of $pooling(GNN(G))$ and the embeddings of $P_w(G)$ to optimize $P_w$. In this case, I believe Figure 1 cannot serve as evidence for Theorem 5 because: 1. Theorem 5 does not establish any relationships between the error and the rank; 2. Figure 1 does not reveal the relationship between the error and $||C(G)||$; 3. Even though $P_w$ can approximate the specific embeddings of $pooling(GNN(G))$ under the guidance of the distance loss function in this case, it does not imply that it can approximate the desired $C(G)$ with the guidance of the downstream task loss function.

---

> > > ### Author Response · Authors · 2024-11-26
> > >
> > > Thanks for your further questions!
> > >
> > >
> > > > ...then set the loss function as the distance between the embeddings of $pooling(GNN(G))$ and the embeddings of $P_w(G)$ to optimize $P_w$.
> > >
> > >
> > > **R 1**: Given a graph $G$, we change the graph and get $G^{'}$. The loss function is **not the distance between $G$ and $P_w(G)$ but $G^{'}$ and $P_w(G)$.**
> > >
> > > - Why? Because here graph embedding of $G^{'}$ is treated as $C(G)$.
> > > - Why $G^{'}$ is treated as $C(G)$? because just as we mentioned previously:
> > >
> > > > In this paper, we argue the powerful capability of graph prompt in manipuating graph data. However, is graph manipulation always helpful for any kind of downstream task (or dataset)? That is an open problem needing to be answered and that might be the reason why graph prompts sometimes do not "work well" in some cases as reported by some empirical study. Figuring out this problem deserve a hundred of new research papers and application studies. To this end, we avoid discussing detailed downstream task. Instead, we focus on one "special" task, the performance of which nearly defininetly relate to data manipuation. This task is:
> > >
> > > > changing part of edges/nodes etc from graph $G$ and get a new graph $G^{’}$, then we have two graphs, $G$ and $G^{’}$. The task target is to find the graph embedding of this manipulated graph $G^{’}$.
> > >
> > > > In this task, $C(G)$ means graph-level embedding of $G^{’}$ (ground truth and we can easily get this ground truth by simply calculating the pooling of the manipulated graph). The we consider how well does graph prompt approximate this ground truth and report the error between them.
> > >
> > > ---
> > >
> > > > ... Theorem 5 does not establish any relationships between the error and the rank;
> > >
> > > **R2**: Theorem 5 said: "...assume at least one layer’s parameter matrix is not full rank...". Therefore, figure 1 considered to reflect the error under different ranks.
> > >
> > >
> > > ---
> > >
> > >
> > > > 2. Figure 1 does not reveal the relationship between the error and $||C(G)||$; 3. Even though $P_w$ can approximate the specific embeddings of $pooling(GNN(G))$ under the guidance of the distance loss function in this case, it does not imply that it can approximate the desired $C(G)$ with the guidance of the downstream task loss function.
> > >
> > > **R3**: I think the reviewer might be trapped in a traditional empirical thinking habit. You might assume that an empirical paper should have one experiment to solve one conclusion. However, this is a theory-oriented paper, we should assume that some Theorems might not be directly "demonstrated" by empirical experiments because these theorems have been strictly demonstrated in our Appendix.
> > >
> > > **Then the question is: How to "demonstrate" the theorem via empirical study?** A mainstream solution is to use several empirical studies to reflect different sides of observations drawn from the theorem.
> > > - As for Theorem 5, we reflect some key observations from it not only in Figure 1, please also check in section 5.3, we have three more experimental figures that try to reflect the theorem comprehensively.
> > > - As for Figure 1: lines 275-281 said:
> > > > Theorem 5 reveals the potential distortion of BG’s shape when the matrix is not full-rank and the model’s expressive power is insufficient. This can lead to an increased distance between BG and the transformation do- main DP (G) of GPF or All-in-One prompts. To To confirm thisjudgment, we conducted a quantitative analysis using numercal methods for the case of non-full rank matrices
> > >
> > >
> > >
> > > We also encourage the reviewer to see our detailed procedure in Appendix C, in which we step by step tell our readers how to conduct Figure 1.

---

> > > > ### Comment · Reviewer_sjg4 · 2024-12-01
> > > >
> > > > Thank you for your response. However, I remain unconvinced by your explanation.
> > > >
> > > > Regarding Theorem 5 and Figure 1, my main concern from my previous comments remains unaddressed: “Even though $P_w$ can approximate the specific embeddings of $pooling(GNN(G))$ under the guidance of the distance loss function in this case, it does not imply that it can approximate the desired $C(G)$ with the guidance of the downstream task loss function.” For this reason, I do not believe that Theorem 5 and Figure 1 provide a meaningful understanding of the power of graph prompts.
> > > >
> > > > Furthermore, I have additional concerns regarding Theorem 7. Specifically, in Theorem 7, you claim that “the eigenvalues of $V$  in datasets often exhibit an exponential decay.” How can you arrive at this conclusion without knowledge of  $C(G)$ ?
> > > >
> > > > Due to the earlier misunderstanding, I re-read the paper and re-evaluated it with a new rating. Overall, I believe this paper requires major revisions and a clearer presentation.

---

> > > > > ### Author Response · Authors · 2024-12-01
> > > > >
> > > > > Dear reviewer,
> > > > >
> > > > > Kindly note that Theorem 5 is an **UPPER Bound** of the error, which means the prompted error is within this upper bound. We give the details in our proof.
> > > > >
> > > > >
> > > > >
> > > > > > Furthermore, I have additional concerns regarding Theorem 7. Specifically, in Theorem 7, you claim that “the eigenvalues of  in datasets often exhibit an exponential decay.” How can you arrive at this conclusion without knowledge...
> > > > >
> > > > > We are conducting a theory analysis, not an empirical one. Let's take an example for you: you might be familiar with math and you might learn such terms like "implicit function", etc, where we usually do not know its detailed format but it won't prevent our analysis to its core quality. Here we do not know the details of C, but it won't prevent our analysis. IF you really understand and read our paper, we do not think it is far-fetched to understand.
> > > > >
> > > > >
> > > > > IF you changed your mind, we thank you. IF not, we thank you too because we understand it is difficult for a non-expert in this area to read such a theory-intensive paper.
> > > > >
> > > > >
> > > > > Kind regards.

---

### Official Review · Reviewer_VdLJ · 2024-11-04

**Soundness:** 3
**Presentation:** 3
**Contribution:** 3
**Rating:** 6
**Confidence:** 3

**Summary:**

The paper presents a theoretical framework for understanding graph prompting, a method of incorporating additional tokens or subgraphs without requiring retraining of pre-trained graph models over various downstream tasks. The authors introduce a comprehensive theoretical framework that establishes formal guarantees on the effectiveness of graph prompts in approximating various graph transformation operations. They derive upper bounds on the errors introduced by these prompts for individual graph, and further extend the findings across different GNN architectures, including both linear and non-linear models. The empirical study supports the theoretical findings and showcases the practical benefits.

**Strengths:**

The paper introduces a comprehensive theoretical framework for graph prompting, significantly advancing the understanding of how and why graph prompts work. Specifically,
- There are some key concepts such as "bridge sets" and "ϵ-extended bridge sets," which are pivotal for understanding how graph prompts function. By establishing the existence of a bridge graph for any given input graph, the authors provide a strong theoretical foundation that justifies the use of graph prompts.
- The paper provides a thorough and systematic analysis of the errors introduced by graph prompts, establishing upper bounds that offer valuable insights into their performance. By deriving specific error bounds, the authors contribute significantly to the theoretical understanding of graph prompting.

The paper is well-structured, featuring clearly defined notations and formulas that significantly contribute to its readability and comprehension

**Weaknesses:**

The paper offers rigorous theoretical analysis related to data operation perspective, upper bound study, etc. In the meantime, it may lack sufficient contextualization regarding how these error bounds apply in real-world scenarios. A more practical interpretation of the results could help bridge the gap between theory and application.

In Section 4, the theorems are based on the assumption of full-rank weight matrices. It would be helpful to investigate how well the assumption holds in practical applications

The experimental section of the paper lacks a comprehensive integration of real-world dataset evaluations within the main text, which limits the visibility and perceived relevance of the findings. By omitting a detailed discussion of these real-world results in the primary analysis, the paper misses an opportunity to contextualize its findings and demonstrate the effectiveness of graph prompting in practical applications. Including these insights in the main text would strengthen the paper’s overall impact

**Questions:**

In the experimental section, the authors utilize GCN and GAT as the primary models. Given that graph transformers have emerged as SOTA models especially in the context of prompting, could you elaborate on your decision to focus on GCN and GAT? How do you believe your findings might differ if applied to graph transformers?

Based on the theoretical study in the paper, what are the next steps you envision for further research in graph prompting? How can practical application benefit from the theoretical framework? It would be good to have the paper linked to real-world applications

---

> ### Author Response · Authors · 2024-11-22
> **Rebuttal to VdLJ**
>
> We truly thank your positive support of this work! According to other reviewers’ scores (8 and 3), your opinion is very important to us because it is just like a ``battleground state of the American Presidential Election`` : ) Below we respond to your questions and suggestions one by one. **Hope they will help encourage your further raising score!**

---

> > ### Author Response · Authors · 2024-11-22
> >
> > > W1: The paper offers rigorous theoretical analysis related to data operation perspective, upper bound study, etc. In the meantime, it may lack sufficient contextualization regarding how these error bounds apply in real-world scenarios. A more practical interpretation of the results could help bridge the gap between theory and application.
> > >
> >
> > **R-1:** Thank you for your insightful comment. We agree that bridging the gap between theory and application is essential for advancing practical understanding and utility. In practice, the design and training of prompts often rely on empirical results and heuristics. Our theoretical analysis provides error bounds that can directly inform and guide prompt design and analysis in real-world scenarios. Specifically:
> >
> > - **Theorems 5 and 8** highlight that rank deficiency in the model’s parameter matrix can lead to information loss in prompts. This means that if a prompt is not performing well, it may be due to the pre-trained model exhibiting a non-full-rank condition—an experience that has not been previously mentioned in this field. Recognizing this issue allows practitioners to focus on the rank properties of their models to retain essential information from prompts.
> > - **Theorem 6** indicates that for datasets of a certain scale, prompts with limited complexity have a theoretical upper bound on performance. This suggests that to achieve better results, we need to increase the complexity of the prompts. This theorem provides practical guidance on determining the necessary complexity level of prompts relative to the dataset size.
> > - **Theorem 7** offers a method to estimate the required size of a prompt through the value of *ϵ*. By calculating *ϵ*, practitioners can infer whether the prompt’s size is sufficient for the dataset in question. If a prompt underperforms on a given dataset, computing *ϵ* helps determine whether the issue stems from theoretical limitations (implying the need for a more complex prompt) or from not yet finding the optimal parameters—thus indicating a need for further training.
> >
> > Kindly note that we only list a small part of useful ideas. We are happy to say that our theory serves as a “spade” and we encourage future researchers to dig for more insightful gold using this spade. These theoretical insights provide practical tools for prompt design and analysis, helping practitioners understand and overcome performance limitations in real-world applications. By applying our error bounds, one can make informed decisions about model adjustments and prompt complexity to achieve desired outcomes.
> >
> > We appreciate your feedback and will incorporate a more detailed discussion of these practical implications in the final version of the paper to enhance the connection between our theoretical results and their real-world applications.

---

> > ### Author Response · Authors · 2024-11-22
> >
> > > W2: In Section 4, the theorems are based on the assumption of full-rank weight matrices. It would be helpful to investigate how well the assumption holds in practical applications.
> > >
> >
> > **R-2:** Thank you for your insightful comment regarding the assumption of full-rank weight matrices in Section 4. We appreciate the opportunity to clarify how this assumption holds in practical applications.
> >
> > 1. **Prevalence of Full-Rank Matrices in Well-Trained Models**: As mentioned in our paper, well-trained models typically contain full-rank weight matrices. Common initialization techniques such as orthogonal initialization and He initialization ensure that the weight matrices start as full-rank. During training, these matrices tend to maintain their full-rank property due to the nature of gradient-based optimization processes.
> > 2. **Expressive Power of Full-Rank Matrices**: Full-rank matrices inherently possess stronger expressive capabilities. Training algorithms aim to optimize the model’s expressive power to capture complex patterns in data. Therefore, it is intuitive and reasonable to assume that the training process favors the retention of full-rank weight matrices to achieve better performance.
> > 3. **Mathematical Justification**: From a mathematical standpoint, the set of non-full-rank matrices has measure zero in the space of all matrices. This is because the determinant function is continuous, and a matrix is singular (non-full-rank) only when its determinant is exactly zero. Consequently, the probability of a randomly initialized matrix being non-full-rank is negligible.
> >
> > **Conclusion**: Under typical conditions and standard training practices, it is reasonable to assume that pre-trained models yield full-rank weight matrices. This assumption is both theoretically sound and practically observed, making it a valid basis for our theorems.
> >
> > 1. **Addressing Non-Full-Rank Cases**: We acknowledge that this assumption may not hold in all scenarios. To account for situations where weight matrices might not be full-rank, we have conducted additional analyses in Sections 3 and 5. These sections focus on cases without the full-rank assumption, providing a more comprehensive understanding of the model’s behavior under different conditions.
> >
> > We will enhance the final version of the paper by including a more detailed discussion on the validity of the full-rank assumption in practical applications, along with empirical evidence and potential limitations. Thank you again for your valuable feedback. We believe that addressing this point strengthens our work and its applicability to real-world scenarios.

---

> > ### Author Response · Authors · 2024-11-22
> >
> > > W3: The experimental section of the paper lacks a comprehensive integration of real-world dataset evaluations within the main text, which limits the visibility and perceived relevance of the findings. By omitting a detailed discussion of these real-world results in the primary analysis, the paper misses an opportunity to contextualize its findings and demonstrate the effectiveness of graph prompting in practical applications. Including these insights in the main text would strengthen the paper’s overall impact
> > >
> >
> > **R-3:** We thank the reviewer for this comment. Kindly note that we also included real-world dataset evaluation in our Appendix (see Appendix B), from which we can find similar observations.
> >
> > ---
> >
> > > Q1: In the experimental section, the authors utilize GCN and GAT as the primary models. Given that graph transformers have emerged as SOTA models, especially in the context of prompting, could you elaborate on your decision to focus on GCN and GAT? How do you believe your findings might differ if applied to graph transformers?
> > >
> >
> > **R-Q1:** We thank the reviewer for this comment.
> >
> > - Kindly note that from the theory perspective, and in the setting of this paper, there is no fundamental difference between GAT and Graph Transformers. The difference is that GAT uses an attention mechanism upon the topological structure and graph transformer uses an attention mechanism upon a complete graph but still with a position mask upon this complete graph to preserve the topological structure for information aggregating. They differ in engineering details but have no significant or fundamental difference in mathematics.
> > - In our theory analysis, we choose GCN and GAT because they are the most classic, simplest, and the most representative models. Graph Transformers have many unnecessary notations for delivering our theory. Therefore, we avoid being trapped in unnecessary math complexity so that we can **clearly present** the nature of graph prompt in theory to our readers, without too many trivial distractions. We believe it is a very important principle for a theory-intensive paper to keep elegant, concise, and simple when reflecting the theory in nature.
> >
> > ---
> >
> > > Q2: Based on the theoretical study in the paper, what are the next steps you envision for further research in graph prompting? How can practical application benefit from the theoretical framework? It would be good to have the paper linked to real-world applications.
> > >
> >
> > **R-Q2:** Thank you for your thoughtful comment and for highlighting the importance of connecting our theoretical work to practical applications in graph prompting. We are proud of this work, as it not only advances the theoretical understanding of graph prompting but also redefines our recognition of previous efforts in the field.
> >
> > We emphasize that the potential and future of graph prompting extend beyond being merely a tuning trick. Instead, we should focus on its capability to learn and implement graph data manipulation strategies. This shift in perspective opens up new avenues for research and practical applications.
> >
> > Regarding the next steps we envision for further research in graph prompting:
> >
> > 1. **Data-Operation Intensive Applications**: We believe that future research should explore how graph prompts can be leveraged to achieve applications that require complex data operations. For instance, designing advanced graph prompts to integrate and reason over multi-source graph databases could significantly enhance data analysis and interpretation.
> > 2. **Cross-Domain Transfer**: Another promising direction is using graph prompts to facilitate cross-domain transfer for graph models. This could enable models trained on one type of graph data to be effectively applied to different domains with minimal adjustments, thereby improving adaptability and efficiency.
> > 3. **Enhanced Graph Prompt Design**: We see potential in developing more sophisticated graph prompts that can capture intricate patterns and relationships within graph data. This could lead to better performance in tasks such as network analysis, recommendation systems, and biological modeling.
> >
> > In our paper, particularly in the Introduction, we have listed several real-world applications where graph prompts are utilized. These include social network analysis, knowledge graph completion, and molecular property prediction. By grounding our theoretical framework in these practical contexts, we aim to bridge the gap between theory and application.
> >
> > We believe that our theoretical contributions provide a solid foundation for these future explorations and can significantly benefit practical applications by offering insights into the capabilities and limitations of graph prompting techniques.
> >
> > Thank you again for your valuable feedback. We are committed to further linking our theoretical findings with real-world applications and will elaborate on these connections in the final version of the paper.

---

> ### Author Response · Authors · 2024-11-28
>
> Dear Reviewer VdLJ
>
> We truly appreciate your time in reviewing this work! We write to ask whether your concerns have been addressed by our previous response. Please do not hesitate to let us know since phase 1 will soon come to a close.
>
> We noticed that you gave us a very high score at Soundness: 3: good | Presentation: 3: good | Contribution: 3: good, but a relatively slightly positive score at Rating: 6: marginally above. May I know if is there anything we can do to further address your concern?
>
>
>
> We humbly wish you could reconsider our work if the mentioned misunderstandings are clarified. According to other reviewer scores (8 and 3, respectively), your potential further raising might significantly help us to fight for our acceptance. We believe our work deserves a more favorable score if your misunderstandings are clarified.
>
> Graph prompts have recently been widely treated as a very promising way at the data level towards more general graph-based AI applications. However, despite its potential, ``the theoretical foundation of graph prompting is nearly empty, raising critical questions about its fundamental effectiveness.`` The lack of rigorous theoretical proof of why and how much it works is more like a **“dark cloud”** over the graph prompting area to go further. **And that is why we are truly proud of this work as it advances the theoretical understanding of graph prompting and redefines previous efforts in the field.**
>
>
> Currently, the research community is filled with too many empirical studies but we urgently need to figure out these foundational theories to support us to go further. Our contributions provide a solid foundation for future explorations and can significantly benefit practical applications by offering insights into the capabilities and limitations of graph prompting techniques.
>
>
> Thanks again for your time. We are warmly looking forward to your letter.
>
> Kind regards

---

> > ### Comment · Reviewer_VdLJ · 2024-12-02
> >
> > Thanks for the detailed rebuttal and additional insights. I will retain my positive score

---

### Author Response · Authors · 2024-11-22
**To All Reviewers**

**Dear Reviewers,**

We would like to express our deepest gratitude for your time and insightful feedback on our manuscript titled "Does Graph Prompt Work? A Data Operation Perspective with Theoretical Analysis." Your constructive comments have been invaluable in enhancing the quality and clarity of our work. We are committed to advancing the field of graph prompting and believe that our revisions have significantly strengthened the paper.

We have carefully considered all the comments and have made substantial revisions to address the concerns raised. We highlight part of the changes in red in our revised manuscript.  Also, we provide more experiments (https://anonymous.4open.science/r/dgpwadopwta/supplement.pdf) and more open discussion (https://anonymous.4open.science/r/dgpwadopwta/OpenDiscussion.pdf) as expected by reviewer ``VdLJ``, both of which can be accessed from our open code project.

We appreciated Reviewer ``VdLJ`` and Reviewer ``sjg4`` for their positive support to our work (with rating scores **6 and 8**, respectively). We also truly appreciated the comment given by Reviewer ``q53a``, in which we believe most of the concerns arose from misunderstandings, and we have taken steps to clarify.


**Significance and Impact of Our Work**

We are truly proud of this work as it advances the theoretical understanding of graph prompting and redefines previous efforts in the field. Our contributions provide a solid foundation for future explorations and can significantly benefit practical applications by offering insights into the capabilities and limitations of graph prompting techniques.
- **Theoretical Foundations:** By establishing rigorous guarantee theorems and deriving upper bounds on data operation errors, we provide essential theoretical foundations that were previously lacking in the field.
- **Practical Implications:** Our work offers practical tools and guidance for prompt design and analysis, helping practitioners overcome performance limitations and make informed decisions about model adjustments.
- **Future Research Directions:** We open new avenues for research, encouraging the development of advanced graph prompts and their application in data-operation-intensive tasks and cross-domain transfer.


**Conclusion and Request for Reconsideration**

We sincerely hope that our detailed revisions and clarifications have addressed all concerns and demonstrated the strength and relevance of our work. We are committed to contributing to the field and believe that our paper offers significant value to both the research community and practical applications.

We kindly request the reviewers to consider our revisions and the efforts made to enhance the manuscript. Your support is crucial, and we hope that you will view our work favorably and consider raising your scores to reflect the improvements made.

Thank you once again for your time and valuable feedback.

Best regards

---

### Author Response · Authors · 2024-12-03
**Letter of Thanks and Withdraw Declaration**

Dear Reviewer **VdLJ**, Reviewer **sjg4**, Reviewer **q53a**, and Area Chair,


Today is a hard day for us, but it is also a day to express our heartfelt thanks. We are grateful to every reviewer for giving their time in this phase and for walking alongside our paper. In the past few days, we were happy to see that all the reviewers clearly recognized the huge contributions and significance of our work to push graph AGI, especially graph prompting, forward.


We also truly appreciate your constructive feedback, from which we realized that the main shortcomings of this theory-intensive paper currently is to further reduce the understanding bar for non-specialists or someone who lacks a basic math background.


**We decided to withdraw our paper, revise it harder, and submit it to the next high-level conference. We hope we can meet you somewhere else shortly and meet you at the next venue.**


Since you might know who we are soon, we kindly hope we can have a chance to cooperate with you in the future. Please go through our webpage to see our latest research and do not hesitate to help us increase our academic impact by spreading, citing, or discussing our work if you like.




Sincerely,

The Authors.

---

### Note · Authors · 2024-12-03

I have read and agree with the venue's withdrawal policy on behalf of myself and my co-authors.